# Interactions between the accumulation of sediment storage and debris flow characteristics in a debris-flow initiation zone, Ohya landslide body, Japan

Fumitoshi Imaizumi[1], Yuichi S. Hayakawa[2], Norifumi Hotta[3], Haruka Tsunetaka[4], Okihiro Ohsaka[1], Satoshi Tsuchiya[1]

[1]Faculty of Agriculture, Shizuoka University, Shizuoka, 422-8529, Japan
[2]Centre for Spatial Information Science, The University of Tokyo, Kashiwa, 277-0871, Japan
[3]Faculty of Life and Environmental sciences, University of Tsukuba, Tsukuba, 305-8572, Japan
[4]Graduate School of Life and Environmental sciences, University of Tsukuba, Tsukuba, 305-8572, Japan

*Correspondence to*: Fumitoshi Imaizumi (imaizumi@shizuoka.ac.jp)

**Abstract.** Debris flows often occur in steep mountain channels, and can be extremely hazardous as a result of their destructive power, long travel distance, and high velocity. However, their characteristics in the initiation zones, which could possibly be affected by temporal changes in the accumulation conditions of the storage (i.e., channel gradient and volume of storage) associated with sediment supply from hillslopes and the evacuation of sediment by debris flows, are poorly understood. Thus, we studied the interaction between the flow characteristics and the accumulation conditions of the storage in an initiation zone of debris flow at the Ohya landslide body in Japan using a variety of methods, including a physical analysis, a periodical terrestrial laser scanning (TLS) survey, and field monitoring. Our study clarified that both partly and fully saturated debris flows are important hydrogeomorphic processes in the initiation zones of debris flow because of the  steep terrain. The predominant type of flow varied temporally and was affected by the volume of storage and rainfall patterns. Fully saturated flow dominated when total volume of storage < 10,000 m³, while partly saturated flow dominated when total of the storage > 15,000 m³. The small-scale channel gradient (on the order of meters) formed by debris flows differed between the predominant flow types during debris flow events. Partly saturated debris flow tended to form steeper channel sections (22.2 – 37.3°), while fully saturated debris flow tended to form gentler channel sections (<22.2°). Such relationship between the flow type and the channel gradient could be explained by a simple analysis of the static force at the bottom of the sediment mass.

## 1 Introduction

Debris flows often occur in steep mountain channels and can be extremely hazardous as a result of their destructive power, long travel distance, and high velocity (Lin et al., 2002; Cui et al., 2011). To lower the hazard of debris flow, field monitoring has been conducted in many torrents in the world including Switzerland (McArdell et al., 2007; Berger et al., 2011b), Italy

(Arattano, 1999; March et al., 2002; Arattano et al., 2012), the United States (McCoy et al., 2010; Kean et al., 2013), and China (Zhang, 1993; Hu et al., 2011). Many of these monitoring activities have been undertaken in the transportation zones of the debris flows, with only a few observations undertaken in their initiation zones, where the unstable sediment start to move (Berti et al., 1999; McCoy et al., 2012; Kean et al., 2013). Thus, factors affecting debris-flow characteristics in the initiation

zone (e.g., temporal changes in the solid fraction) is still unclear.

Debris flows can be classified into various types according to their flow dynamics, solid fractions, and material types (Coussot and Meunier, 1996; Hungr, 2005; Takahashi, 2007). Multiple flow types appear even in the same torrent (Imaizumi et al., 2005; Okano et al., 2012). However, in situ classification of debris flow type is mainly based on monitoring results obtained in the transportation zone. Field monitoring conducted at multiple sites along a debris flow torrent have revealed that

flow characteristics (e.g., solid fraction and boulder size) and discharge change as the flow migrates downstream and is affected by erosion and deposition (Takahashi, 1991, Berger et al., 2011b; Arattano et al., 2012). Thus, flow characteristics in the transportation zone possibly differ from those in the initiation zone. In the initiation zone, flows containing an unsaturated layer have also been observed during field monitoring (Imaizumi et al., 2005; McArdell et al., 2007; Imaizumi et al., 2016b). Such information is needed to explain the sequence of debris flow processes from initiation in the headwaters to termination

at the debris flow fan.

The topography along debris flow torrents reflects the characteristics of a debris flow event (Whipple and Dunne, 1992; Coussot and Meunier, 1996; Imaizumi et al., 2016a). The curvature of the terrain can be used to determine the relationship between shear stress and shear strength (Staley et al., 2006). Many observations of the relationships between topography and flow characteristics have been made in the lower part of the debris flow torrents, while such findings are limited in the debris-

flow initiation zone. In many cases, the channel gradient in the initiation zone is greater than 20° (VanDine 1985; Pareschi et al. 2002; McCoy et al., 2013; Hürlimann et al., 2015). In terms of debris flows caused by the transportation of loose sediment on the floor of a valley, both the volume and grain size of debris flow material (e.g., channel deposits and talus slope) change over time in association with the sediment supply from hillslopes, as well as the evacuation of sediment by debris flows and fluvial processes (Bovis and Jakob, 1999; Imaizumi et al., 2006; Gregoretti and Dalla Fontana, 2008; Berger et al., 2011a;

Theule et al., 2012). Although some previous studies have investigated the relationship between characteristics of the debris flow material and initiation condition of a debris flow (Bovis and Jakob, 1999; Jakob et al., 2005; Schlunegger et al., 2009; Chen et al., 2012; Theule et al., 2012), only a few have considered the relationship between flow material and flow characteristics (Kean et al., 2013; Imaizumi et al., 2016b). The difficulty in monitoring debris flows in steep and dangerous initiation zones has prevented the collection of the field data needed to clarify the relationship between flow characteristics

and flow material. In addition, the value of topographic indexes (e.g., slope gradient) are variable depending on the scale of the grid size used in the GIS analyses, because factors affecting the indexes are different among the scales of topography (Schmidt and Andrew, 2005; Loye et al., 2009; Pirotti and Tarolli, 2010; Drăguţ and Eisank, 2011). Thus, appropriate grid size for the analysis should be understood prior to discuss relationship between accumulation condition of flow material and flow characteristics.

In the debris-flow initiation zone at the Ichinosawa catchment within the Ohya landslide, Japan, field monitoring has been undertaken since 1998 (Imaizumi et al., 2005; Imaizumi et al., 2006). This site is suitable for monitoring because of the high debris-flow frequency (about three or four events per year) that occur due to the mobilization of storage (i.e., talus cone and channel deposits) around the channel. In addition, detailed topographic measurements by terrestrial laser scanning (TLS) have been undertaken periodically since 2011 (Hayakawa et al., 2016).

The overall aim of this study is to clarify interaction between debris flow type (i.e., fully and partly saturated flows) and the accumulation conditions of storage (i.e., slope gradient and volume of storage). Our specific objectives were to explain the relationship between slope gradient and sediment transport type by a simple analysis of the static force, clarify effects of the accumulation conditions of storage and rainfall pattern on the debris flow type, find out representative slope gradient of geomorphic units (i.e., rock slope, talus slopes, and the channel) based on analysis of digital elevation models (DEMs) with various grid size, and clarify effects of the debris flow type on the channel morphology.

## 2 Slope gradient and type of sediment transport

In this section, we describe our analysis of the balance of static force at the bottom of a sediment mass to assess the relationship between slope gradient and the type of sediment transport, which is the essential to understand observation results and GIS analyses in this study. Shear stress at the bottom of a sediment mass needs to exceed shear strength for the sliding of a stable debris mass (Takahashi, 1991; Prancevic et al., 2014; Imaizumi et al., 2016c). Similarly, shear stress needs to exceed shear strength at the bottom of a traveling sediment mass for the continuity of travel (Takahashi, 1991; Watanabe, 1994). Under the assumption that cohesion is negligible, shear stress $\tau$ and shear strength $\tau_r$ at the bottom of sediment mass can be given as follows:

$$\tau = \{(1 - \eta_w)[(1 - n)\gamma_s + nS\gamma_w] + \eta_w[(1 - n)\gamma_s + n\gamma_w]\}\sin\alpha, \tag{1}$$

$$\tau_r = \{(1 - \eta_w)[(1 - n)\gamma_s + nS\gamma_w] + \eta_w[(1 - n)(\gamma_s - \gamma_w)]\}\cos\alpha\tan\phi, \tag{2}$$

where $\eta_w$ is the ratio of the depth of the saturated zone ($h_w - z_1$) to the depth of the sediment mass ($h - z_1$), $h_w$ is the height of the water table, $z_1$ and $h$ are the height at the bottom and surface of the sediment mass, respectively, $n$ is the porosity of sediment, $\gamma_s$ is the force of gravity acting on a unit volume of sediment, $\gamma_w$ is the force of gravity acting on a unit volume of water (or interstitial water for debris flow), $S$ is the degree of saturation in the unsaturated zone, $\alpha$ is the slope gradient, and $\phi$ is the effective internal angle of friction (Fig. 1). Note that $\alpha$ is the slope gradient of the potential sliding surface when we consider the initial movement of the stable mass, whereas it is the gradient of the surface topography when we consider the migration of the traveling sediment mass. The $n$ in the moving sediment mass is similar to that of stable sediment when dispersion of the sediment particles is not significant, while $n$ in the moving sediment mass greatly exceeds that of the stable sediment when the particle dispersion associated with collision among the particles is significant (e.g., fully saturated debris flow). Based on Eqs.(1) and (2), the critical condition for the movement of sediment mass ($\tau_r = \tau$) can be given as follows (Imaizumi et al., 2016c):

$$\frac{\tan\alpha}{\tan\phi} = \frac{(1-\eta_w)[(1-n)\gamma_s + nS\gamma_w] + \eta_w[(1-n)(\gamma_s - \gamma_w)]}{(1-\eta_w)[(1-n)\gamma_s + nS\gamma_w] + \eta_w[(1-n)\gamma_s + n\gamma_w]}.$$

(3)

Based on the $\eta_w$, three typical sediment transport types were considered: fully unsaturated ($\eta_w = 0$), partly saturated ($0 < \eta_w < 1$), and fully saturated ($\eta_w = 1$). In case of $\eta_w = 0$, Eq. (3) is expressed as:

$\tan\alpha = \tan\phi$, (4)

If the slope gradient $\tan\alpha$ exceeds $\tan\phi$, the sediment mass can move without any saturation (Fig. 1a). In other words, Eq. (4) expresses the lowest boundary of the slope gradient for the movement of the fully unsaturated sediment mass (hereafter referred to as $\alpha_1$). There are several types of sediment transport in fully unsaturated conditions, such as rockfall, dry granular flow, and dry ravel, which is the gravitational transport of ground surface materials by bouncing, rolling, and sliding (Carson,

1977; Dorren, 2003; Gabet, 2003). Although physical mechanisms between individual particles (i.e., rockfall and dry ravel) and flows of grains (i.e., dry granular flow) are different in terms of the interactions among particles, the slope gradient of talus slopes, which are formed by fully unsaturated sediment transport processes, are usually similar or slightly smaller than $\phi$ regardless of the transport type (Kirkby and Statham, 1975; Carson, 1977; Mangeney et al., 2007). Field surveys and laboratory experiments showed that pyroclastic flow, which is a fluid composed of a mixture of air and particles, sometimes reaches

terrain much smaller than $\phi$ (Yamashita and Miyamoto, 1993; Takahashi and Tsujimoto, 2000). Thus, the pyroclastic flow does not satisfy Eq. (4). Although the pyroclastic flow is an important sediment transport process in areas with many fine particles (i.e., volcanic areas), we did not consider that in this study because we focused on sediment transport processes in areas in which gravel, cobbles, and boulders dominated the debris flow material.

The relationship between slope gradient and the volumetric sediment concentration $(1 - n)$ in a saturated sediment mass is

given by substituting 1 into $\eta_w$ in Eq. (3):

$$\tan\alpha = \frac{(1-n)(\gamma_s - \gamma_w)}{(1-n)\gamma_s + n\gamma_w}\tan\phi.$$

(5)

When we apply Eq. (5) to the debris flow, porosity $n$ can be expressed as $1 - C$ using solid fraction $C$. The $n$ in the moving sediment mass becomes larger than the $n$ of storage when sediment particles start to disperse by collision with other particles

(Hungr, 2005; Takahashi, 2014). By the transformation of Eq. (5), relationship between the solid concentration in the steady-state flow (called equilibrium concentration) and the slope gradient is obtained (Takahashi, 1991; Egashira et al., 2001, Takahashi, 2014):

$$C = \frac{\gamma_w\tan\alpha}{(\gamma_s - \gamma_w)(\tan\phi - \tan\alpha)},$$

(6)

An equation of the same structure of Eq. (6) can be obtained through the ratio between the basal bed shear stress and the basal normal stress that have a similar structure to the ration between shear stress and strength (Lanzoni et al., 2017). Equation (5)

just considers shear strength and shear stress at the bottom of the sediment mass, but does not consider the conditions of the fluid of the sediment mass. Thus, the sliding of the sediment mass without any fluid and the plug flow, of which the upper layer in the sediment mass is not fluid, also satisfies Eq. (5). By substituting the porosity of the storage into $n$, Eq. (5) expresses the slope gradient needed for the entrainment of a fully saturated sediment mass, of which solid fraction is same as that of storage (hereafter referred to as $\alpha_2$).When the slope gradient of the terrain ranges between $\alpha_1$ and $\alpha_2$, the transportation of a partly saturated sediment mass occurs (Fig. 1b) (Watanabe, 1994; Imaizumi et al., 2005; Imaizumi et al., 2016c). Partly saturated sediment transport has also been observed in channels gentler than $\alpha_2$ (i.e., channel gradient < 10°) (McArdell et al., 2007; McCoy et al., 2010; Okano et al., 2012). However, the appearance of partly saturated sediment transport in such gentler channels is generally limited at the front of a surge. In torrents gentler than $\alpha_2$, the existence of surface flow over debris flow material is theoretically required for the initiation of debris flow (Fig. 1c) (Takahashi, 1991; Imaizumi et al., 2016c). Once sediment starts to move, it spreads throughout the flow (Fig. 1d). Thus the volumetric solid concentration is lower than that in sediment storage $(1 - n)$.

The explanations above (Eqs. (5) and (6)) are also applicable to the relationship between the solid fraction of the debris flow and the channel gradient formed by erosion and deposition during passage of the debris flow (Takahashi, 1991; 2014). If the amount of water in the sediment mass from the upper channel reaches is constant, the slope gradient of the terrain approaches the gradient given by the substitution of $\eta_w$, $n$, and $S$ of the sediment mass into Eq. (3) as a result of deposition and erosion.

In this study, we call any partly and fully saturated sediment transport processes as partly and fully saturated debris flows, respectively. Although some sediment transport processes at our monitoring site were not typical debris flows, such as those composed of a mixture of sediment and water (Takahashi, 1991; Coussot and Meunier, 1996), we believe the processes at our study site directly or indirectly contributed to the initiation of debris flow. Field monitoring in many debris flow torrents, including Ohya landslide, showed that debris flows on debris deposits laid on the steep channels usually initiate by runoff as a surficial erosion (e.g., Coe et al., 2008; Gregoretti and Dalla Fontana, 2008; Degetto et al., 2015; Imaizumi et al., 2016b). In such cases, because sediment transport was affected by the hydrodynamic forces (Gregoretti, 2000; Gregoretti and Dalla Fontana, 2008; Prancevic et al, 2014), solid concentration of the debris flow does not satisfy Eq. (5) at least in the initial stage of the debris flow. In addition, above discussion does not consider dynamic force in the flow. Therefore above models cannot strictly explain all sediment transport processes in the debris flow torrents. In this study, however, we approximated such complex flow conditions by above simple static models as with Takahashi (2014) and Prancevic et al. (2014) in order to figure out overall interactions between sediment transport type and slope gradient in the debris flow initiation zone.

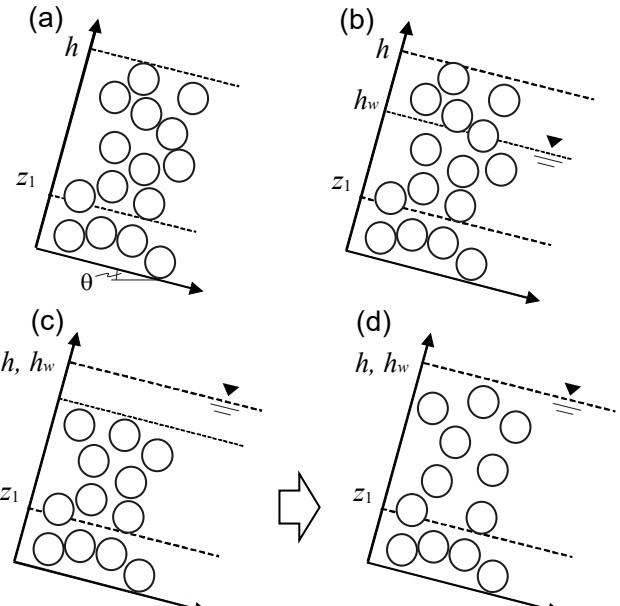

**Figure 1: Schematic diagram of sediment transport types: (a) dry (fully unsaturated), (b) partly saturated, (c) surface flow over sediments, which triggers fully saturated sediment transport, (d) fully saturated sediment transport. The $h$, $h_w$, and $z_1$ in the figure indicate heights at surface of the sediment mass, water table, and bottom of the sediment mass, respectively.**

## 3 Study site

We conducted field monitoring within the Ohya landslide in the Southern Japanese Alps (Fig. 2). The Ohya landslide has an estimated total volume of 120 million m³, and was initiated during an earthquake in A.D. 1707 (Tsuchiya and Imaizumi, 2010). The climate at the site is characterized by a high annual precipitation (about 3,400 mm) (Imaizumi et al., 2005). Heavy rainfall (i.e., total rainfall > 100 mm) occur during the rainy season from June to July and the autumn typhoon season (from August to October). The geological unit is Tertiary strata, which is composed of well-jointed sandstone and highly fractured shale. Unstable sediments have been supplied from outcrops into the channels in the landslide scar and have affected the initiation of debris flows since the original failure.

Almost all of debris flows in the Ohya landslide occur in the upper Ichinosawa catchment (Imaizumi et al., 2005). The total length of the channel is ≈650 m and the south-facing catchment (1,450 – 1,905 m a.s.l) has an area of 0.22 km². Most of the basin is characterized by high and steep slopes (40 – 65°). Seventy percent of the slope is scree and outcrop, whereas the remaining 30% is covered with forest, shrubs, and tussocks. Anthropogenic influences are absent in the catchment because of the harsh environmental conditions.

Unconsolidated debris, ranging from sand particles to boulders (Imaizumi et al., 2016c), is located in the channel bed and talus cones, and is the source of debris flow material (Imaizumi et al., 2006). The thickness of debris deposits, including large

boulders (> 1 m), exceeds several meters in some sections. Freeze-thaw that promotes dry ravel and rockfalls are the predominant sediment infilling of the channels (Imaizumi et al., 2006). Most of the channel was covered by sediment when a large volume of storage accumulated in the upper Ichinosawa, while bedrock exposed in some channel sections when the storage volume is low. Volume of storage displays seasonal changes caused by sediment supply from hillslopes in winter and early spring, and the evacuation of storage due to the occurrence of debris flow in summer and autumn (Imaizumi et al., 2006). The stored sediment has never been completely eroded by debris flows. Changes in the volume of storage at longer time scales (several years) occurs by the timing of large debris flows with a volume > 15,000 m$^3$ that drastically decrease volume of storage (Imaizumi et al., 2016c).

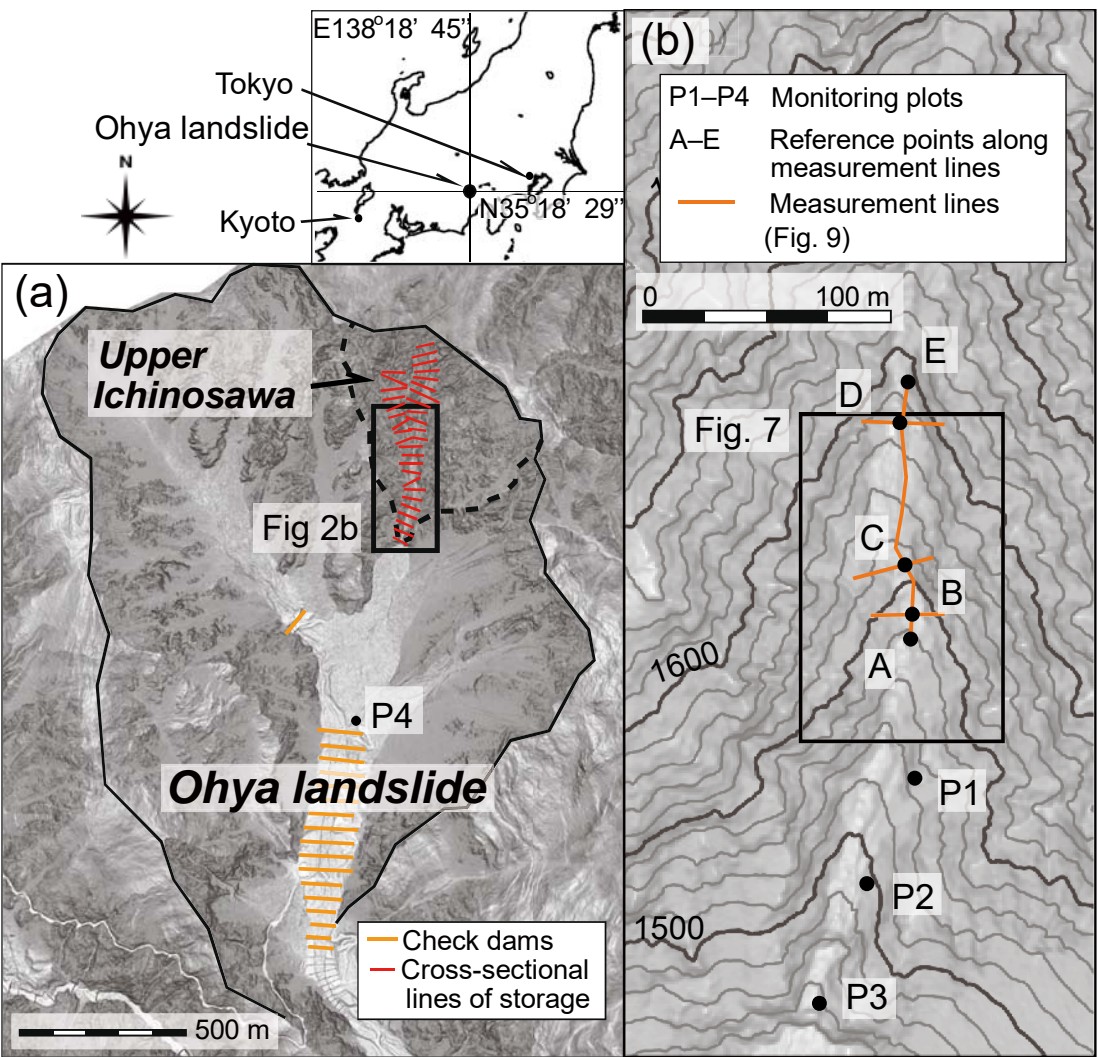

**Figure 2: Map of the Ohya landslide and upper Ichinosawa catchment: (a) Ohya landslide, (b) Ichinosawa upper catchment. Gentler and steeper terrains are expressed as light and dark colors, respectively. Cross-sectional lines used for estimation of volume of the storage are shown as red lines in Figure 2a.**

## 4 Methodology

### 4.1 Debris-flow Monitoring

The monitoring system was installed in the study site in spring 1998 and consisted of a rain gauge, ultrasonic sensors, water pressure sensors, and a video camera (Imaizumi et al., 2005). The video camera, which provided motion images of debris flows, was initially monitored at P4, then moved to P2 in 2000 (Fig. 2). During the period of 1998 to 2001, the video camera was programmed to take 0.75 s clips of video every 5 min. In April 2001, this interval was shortened to 3 min to capture flow characteristics in more detail. In addition, continuous monitoring cameras, of which recording was initiated by wire motion sensors installed at several cross-sections in the channel, were initially installed at P1 in 2003, then at P3 and P2 in 2004 and 2005, respectively. The video camera system sometimes failed to capture debris flows because of their mechanical problems due to harsh site conditions. We identified timing of such debris flows based on changes in the topography observed by periodical photography with an interval of appropriately one week. Surface velocity of debris flows at 1 second intervals were obtained from time required for boulders on the flow surface to pass through fixed channel sections (2.0–5.0 m length) on the video images. The surface velocity provided by the video image analysis does not represent the flow depth averaged velocity. The flow depth averaged velocity was estimated from surface velocity multiplied by 0.6, based on the velocity profile throughout the flows on movable beds obtained from a physically based model by Takahashi (1977, 2014). Flow depth of debris flows at 1 second intervals were also obtained from the video image analysis by reading level of the flow surface. Analysis points of the flow depth were set within the channel sections where changes in the channel bed topography attributed to occurrence of debris flows were minimum. Changes in cross-sectional area of flow were calculated from changes in flow depth and cross-section measurement of the channel topography. Discharge of debris flows were estimated from the cross-section area multi-plied by mean velocity of all layers.

Imaizumi et al. (2006) classified the flow phases during debris flow events into two primary types: flows that made of mainly muddy water and those made of mainly cobbles and boulders. The former flows are turbulent and are characterized by black surfaces due to high concentrations of silty sediment sourced from shale in the interstitial water (Fig. 3a). Because the matrix of boulders is filled with interstitial water, this flow is considered fully saturated (Fig. 1d). In contrast, the flow surface of the second type of flow is unsaturated, because muddy water is not identified in the matrix of the flow surface (Fig. 3b). Therefore, this type of flow is considered partly saturated (Fig. 1b). We visually identified temporal changes in the flow type during debris flow events from video images based on the existence of interstitial water on the flow surface.

Time-lapse cameras (TLCs; GardenWatchCam, Brino, Taipei City, Taiwan) were installed around P1 and P2 in April, 2013, as a backup for video camera monitoring. The number of cameras (1 to 6) differed among the various measurement periods. The intervals used for capturing images were also different among the measurement periods and for the various cameras, with a range from 1 s to 10 min.

To identify the arrival of debris flow, a semiconductor type water pressure sensors, which monitored hydrostatic pressures up to 49 kPa with an accuracy of $\pm$ 3%, were also placed in holes dug in the bedrock of the channel bed at P3. Because water pressure sensors were sometimes washed away by debris flow, ultrasonic sensors with a measuring range from 120 to 600 cm (accuracy of $\pm$ 0.4%) were installed to monitor the surface height of debris flows as the backup of water pressure sensors. Because the logging interval, which was set at 1 min for both types of sensors, was similar or longer than duration of debris flow surges (generally 20 s to 1 min), these sensors were used not to calculate the discharge, but to identify occurrence of debris flows. Water pressure and flow height monitored by these instruments showed intense and abrupt changes during the passage of debris flows, whereas such changes were absent during rainfall-runoff events without occurrence of debris flow.

Precipitation has been measured with logging interval of 1 min using a tipping bucket rain gauge (0.5 mm for one tip) located in an open area at P1 (Fig. 2b). We separated rainfall events by intermissions longer than 6 hr.

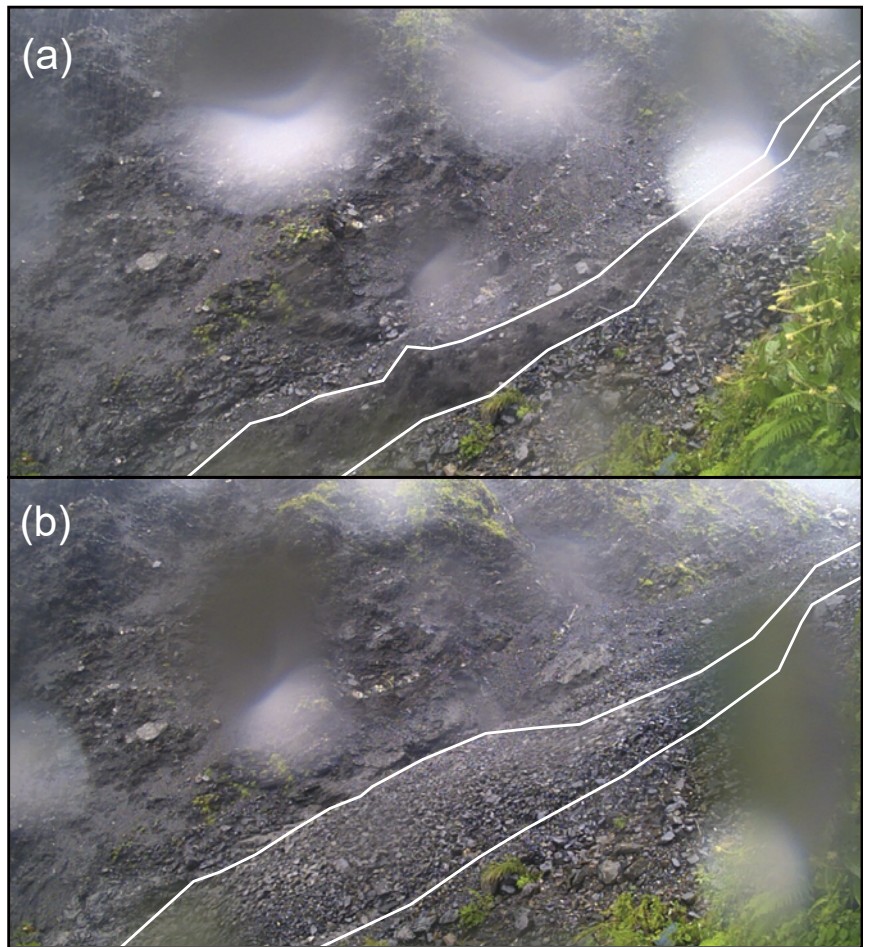

**Figure 3: Images of fully and partly saturated debris flows captured by a time-lapse camera (TLC) at plot P2 in Fig. 2. (a) Fully saturated debris flow captured 9 September 2015, 8:41 (LT). (b) Partly saturated debris flow captured at 9 September 2015, 7:43 (LT). Cobbles and boulders covers flow surface of the partly saturated debris flow (Imaizumi et al., 2016b).**

## 4.2 Topographic Data Obtained by TLS

To clarify temporal changes in the micro channel bed topography associated with occurrence of debris flows, a TLS unit (GLS-1500 Topcon Co., Tokyo, Japan) was used to measure the topography of valley axis (Fig. 2) (Hayakawa et al., 2016). Based on the specifications, the maximum measurable distance of the TLS unit is 500 m (for target objects with a 90% reflectance),

10 with accuracies of 6" for angle and 4 mm for distance (at 150 m range). From November 2011, field measurements were undertaken in spring, summer, and autumn for 4 years (Table 1). For each measurement, the scanner was set at two positions and was correctly leveled with its internal tilt sensor (giving an angle accuracy of 6''). One was at the downstream side of the target area (P1 in Fig. 2), where the land had been relatively stable for many years. The other at the upstream side of the target

area (around point D in Fig. 2). The location of this scan position varied with time because it was in the valley bottom where the topography changed due to erosion and deposition by debris flows. The two point clouds measured from different scan positions were registered using at least five reference targets placed between the two scan positions with accuracies of 0.5–6 mm. The point spacing of cloud points ranged from 0.038 to 0.130 m (Table 1), thus the point densities ranged from 59.5 to 689.9 pts m$^{-2}$ (average 249.6 pts m$^{-2}$). The geographic coordinates (Japan Plane Rectangular CS VIII, EPSG: 2450) of two targets, selected as georeference points, were measured with global navigation satellite system (GNSS) receivers (GeoXH 6000, Trimble, Sunnyvale, CA, USA or GRS-1, Topcon Co., Japan; accurate to a range of 10–63 mm in XY or Z). The baseline solution was calculated using data from GNSS base stations of GEONET, the Japanese GNSS network operated by the Geospatial Authority of Japan. The overall uncertainties of the point clouds, including scanning, registration, and georeferencing by GNSS, were considered in the order of centimeters to a decimeter (Hayakawa et al., 2016).

**Table 1: Date of TLS survey**

| Year | Date of survey | Number of debris flow after previous scanning | Date of last debirs flow event[*1] | Aveage point spacing of the cloud points (m) |
|---|---|---|---|---|
| 2011 | November 11 | - | October 14 | 0.075 |
| 2012 | May 14 | 0 | - | 0.077 |
| | August 23 | 2 | June 22 | 0.056 |
| | November 21 | 3 | September 30 | 0.038 |
| 2013 | May 10 | 0 | - | 0.078 |
| | August 16 | 0 | - | 0.040 |
| | November 19 | 2 | September 15 | 0.057 |
| 2014 | May 16 | 0 | - | 0.094 |
| | August 17 | 1 | August 10 | 0.066 |
| | November 28 | 2 | October 5 | 0.085 |
| 2015 | May 15 | 0 | - | 0.094 |
| | August 23 | 4 | August 17 | 0.083 |
| | December 4 | 1 | September 9 | 0.130 |

*1 Date of the last debris flow event are not listed when debris flow had not occurred in the year , because the topography was possibly affected by the winter sediment supply rather than the last debris flow in the previous year.

After manually eliminating noise and unnecessary points, the point clouds obtained by the TLS measurement were converted into DEMs using a simple liner interpolation by triangulated irregular network (TIN) and resampling, because vegetation cover was absent in the site. The resolution of the DEMs was set at 10 cm, which sufficiently covers the average point density of the point clouds. Areas with sparse point clouds (approximately <100 point m$^{-2}$), vegetated areas, and areas affected by the combination of various processes (e.g., talus slopes with the lower part eroded by stream flow) were excluded from the analyses.

The extent of typical geomorphic units in the TLS survey area, including three rock slopes, three talus slopes, and a channel around the monitoring plot P1, was mapped by field surveys (Fig. 4). Distribution of the slope gradient with in these geomorphic units were calculated from TLS DEMs with various grid sizes. The mapping was conducted at the same time as

each TLS survey because the area changed over time due to the sediment supply from outcrops and transportation of sediment by debris flows.

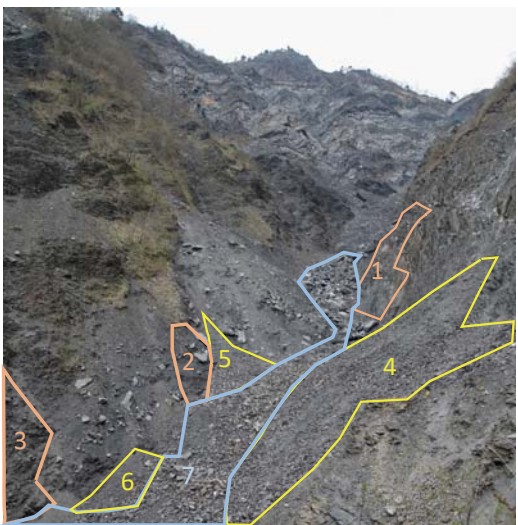

**Figure 4: Photograph of the typical geomorphic units seen from P1. The image shows the area of rock slopes (surrounded by red lines), talus slopes (surrounded by yellow lines), and a channel (surrounded by blue lines), for which the slope gradient was calculated from DEMs with various grid sizes. The numbering of the area corresponds to that used in Fig. 7.**

### 4.3 Estimation of debris-flow volumes

We estimated the volume of storage from photographs taken periodically at sites P1 and P4 (Fig. 2) (Imaizumi et al., 2016a). We could not use data obtained by periodical TLS, because the target area of the TLS was a fraction of the storage zone. Photographs from site P4 cover the entire study site, whereas those from P1 focus on channel deposits at the bottom of the incised main channel, which is shaded in photographs from P4. By comparing these photographs with catchment topography and ortho photographs, which are obtained by airborne laser scanning (ALS) in seven periods (2005, 2006, 2009, 2010, 2011, 2012, and 2013), the area covered by storage (i.e., channel deposits and talus slopes) in each photograph periods was mapped on GIS. The bedrock topography in the upper Ichinosawa catchment was estimated from terrains by ALS in the periods when sediment storage was almost absent (i.e., in 2011 and 2012). Thirty-three cross-sectional areas of the storage with spacing of 25 m along the channels (24 cross-sections along the main channel and 9 cross-sections along a tributary) were calculated from the bedrock topography estimated from terrains by ALS and the location of both ends of the storage along the cross-sections on the GIS storage map (Fig. 2a) under the assumption that surface topography of the storage was an inclined line connecting both ends of the storage. Total volume of the storage was calculated by sum of the cross-sectional area multiplied by the spacing of cross-sections (25 m). The root mean squared error (RMSE) of the procedure, which was obtained from comparison of the volumes of storage by the procedure and that by subtraction of ALS DEMs by estimated bedrock topography, was 5361 $m^3$ (Imaizumi et al., 2016a). This RMSE is larger than the volume of small debris flows (<2000 $m^3$) but below the sediment

supply volume in each year (>10,000 m$^3$). This error may be similar or slightly larger than the errors in field surveys by Hungr et al. (2008), which estimated volume of the eroded sediment 17% apart from that of the deposited sediment.

## 5 Results

### 5.1 Debris flow monitoring

From 1998 to 2015, 59 debris flows, which went through or terminated in the reach between the sites P1 to P3, were observed by video cameras, field surveys, and water pressure sensors in the upper Ichinosawa catchment. In addition, the occurrence of some other debris flows, which terminated above site P1, were found by periodic photography and field surveys. We analyzed video images of seven flows that were clearly captured by video cameras. In addition, we analyzed photographs of one debris flow taken by TLCs with an interval of 1 s (Table 2). We failed to obtain clear video images of other debris flows because of

darkness at night and fog. Rainfall threshold for the occurrence of debris flow can be given by the 10-min rainfall intensity in the upper Ichinosawa catchment (Imaizumi et al., 2005). Comparison between duration of the rainfall event and the maximum 10-min rainfall intensity indicates that the rainfall threshold separating rainfall events with and without debris flows can be given by 5.0 mm 10-min$^{-1}$ in case of rainfall duration ≥10 h (Fig. 5). In case of rainfall duration <10 h, the threshold can be obtained as a linear line in the double logarithmic plot (Fig. 5), as with that in other debris flow torrents (Coe et al., 2008;

Badoux et al., 2008; Theule et al., 2012). Most of the rainfall events triggering debris flows in the study site can be classified into two groups: long-lasting rainfall events with high value of total rainfall depth, caused by typhoons and stationary fronts (rainfall duration >5 h and total rainfall depth >50 mm), and short-duration convective rainfall events characterized by high intensity (rainfall duration <5 h, total rainfall depth <50 mm; Table 2).

As noted above, two types of flow have been observed in the Ohya landslide: flow mainly composed of cobbles and boulders

(partly saturated flow) and flow mainly composed of muddy water (fully saturated flow). The duration of each flow phase varied between the events. For example in the event of 5 August 2008, 88% of debris flow surges (percentage respect to the total event duration) was composed of partly saturated flow, while in the event of 30 August 2004 (Fig. 6), 90% in time of the phenomenon was composed of fully saturated flow. The proportional duration of the partly saturated debris flow in overall debris flow surges was generally high during convective rainfall events with high rainfall intensity and short rainfall duration

(Fig. 5, Table 2). The proportional duration of the partly saturated debris flow also had a weak positive relationship with the volume of storage estimated from periodical photography (R$^2$ = 0.37, p-value = 0.06; Fig. 7).

**Table 2: Details of the rainfall and flow types examined using video camera and time-lapse camera (TLC) images during the debris flow events analyzed in this study.**

| Date | Total rainfall (mm) | Maximum 10-min. rainfall intensity (mm/10min.) | Rainfall duration | Rainfall type | Type of camera | Location of camera | Proportional duration of partly saturated flow in overall debris flow surges |
|---|---|---|---|---|---|---|---|
| 30 August 2004 | 281.5 | 7.5 | 55 h 37 min. | Typhoon | Video | P3 | 0.10[*a] |
| 19 July 2006 | 195.0 | 5.5 | 57 h 46 min. | Stationary front | Video | P2 | 0.65 |
| 6 September 2007 | 501.5 | 9.5 | 37 h 28 min. | Typhoon | Video | P2 | 0.49[*a] |
| 5 August 2008 | 41.0 | 13.5 | 4 h 55 min. | Convective | Video | P2 | 0.88 |
| 24 July 2010 | 21.0 | 13.0 | 51 min. | Convective | Video | P2 | 1.00 |
| 30 September 2012 | 86.0 | 6.5 | 7 h 29 min. | Typhoon | Video | P3 | 0.23[*a] |
| 6 August 2015 | 44.5 | 21.0 | 1 h 19 min. | Convective | Video | P1 | 0.99 |
| 9 September 2015 | 203.0 | 8.0 | 36 h 37 min. | Typhoon | TLC | P1 | 0.42 |

[*a] Latter half of the debris flow event was not analyzed because of the darkness affected by the sunset. Therefore, latter half of the event is not included in this data.

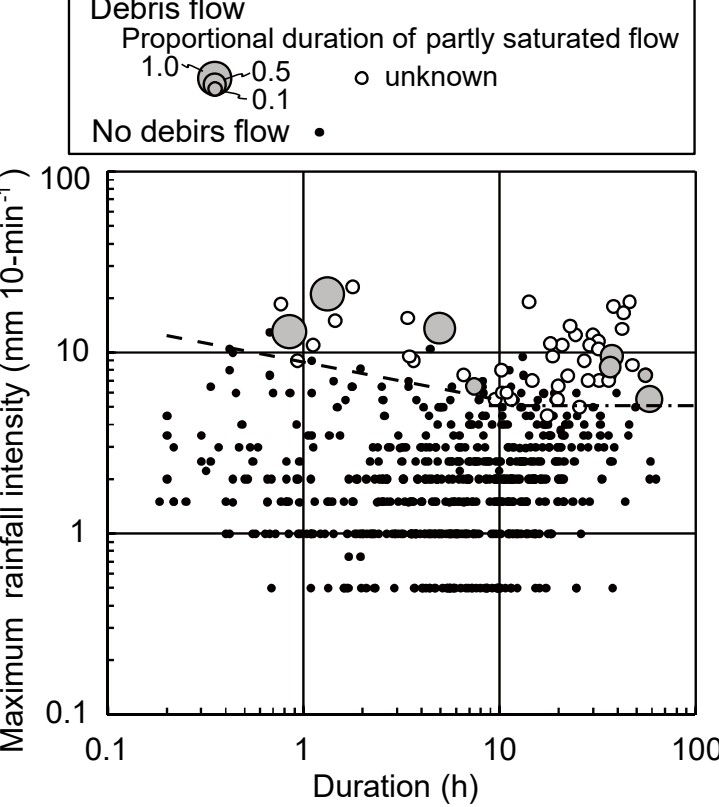

**Figure 5: Comparison between rainfall duration and maximum 10-min rainfall intensity of rainfall events with total rainfall depth >2.0 mm and rainfall duration >10 min in the period from April 1998 to September 2015. Rainfall events with and without debris flows were potted using different markers. Size of the plot for debris flow events expresses proportional duration of partly saturated flow in overall debris flow surges. The rainfall intensity was calculated from rainfall data with a logging interval of 1 min. Rainfall events without 1-min interval data were not plotted. Dashed line, which can be expressed as Intensity = 8.86 (Duration)$^{-0.21}$, indicates lower limit of the rainfall condition needed for occurrence of debris flows when rainfall duration < 10 hr. Dash-dot line indicates the maximum rainfall intensity equals to 5 mm per 10-min, which is the rainfall threshold for occurrence of debris flow when the rainfall duration ≥ 10 hr.**

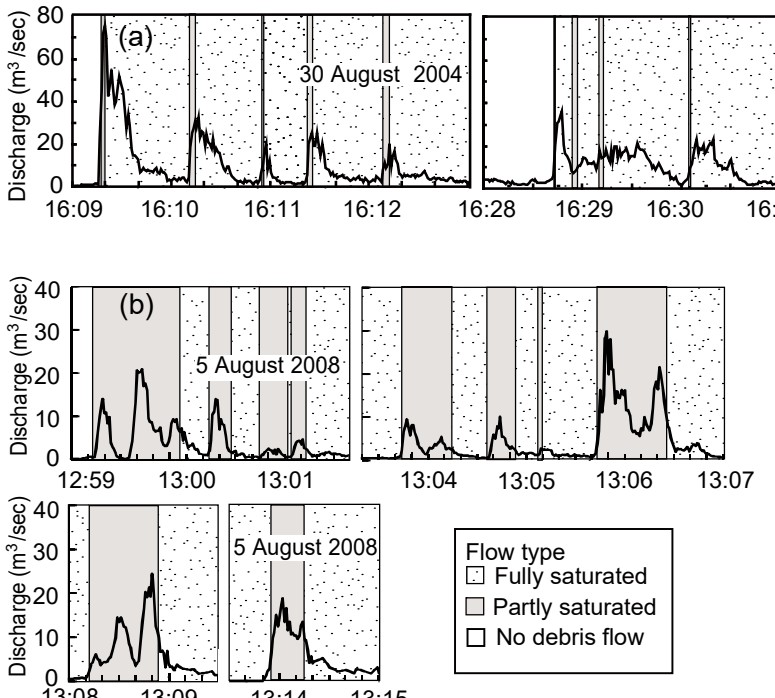

**Figure 6: Hydrograph and the duration of each flow type during debris flow events. (a) Debris flow on August 30, 2004, mainly consisting of fully saturated flow. (b) Debris flow on August 5, 2008, mainly consisting of partly saturated flow.**

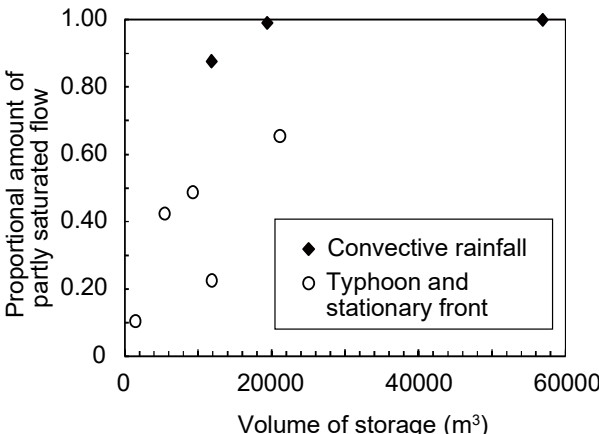

Figure 7: Comparison between the volume of storage and the duration of partly saturated flow as a proportion of the overall debris flow surges. Rainfall duration and total rainfall depth of the convective rainfall are <5 h, and <50 mm respectively, while those of typhoon and stationary fronts are >5 h, and >50 mm respectively.

## 5.2 Slope gradient of geomorphic units

Mapping of the three geomorphic units (rock slopes, talus slopes, and the channel) in the TLS survey area showed that the size of each unit changed with time due to sediment supply and transport processes. Some talus slopes completely disappeared in several periods because of erosion by debris flows. Slope gradient was calculated using 3x3 neighbourhood around each cell in the DEM with various grid size obtained by TLS (Horn, 1981; Fig. 8). On the slope gradient map with a grid size of 0.1 m, the slope gradient is spatially variable attributable to boulders at a scale on the order of decimetres (Fig. 8). Step-pool-like topography with a height of several meters formed by large boulders (>1.0 m) and bedrock morphology was visible in the channel on the slope gradient maps with a grid size of ≥1.0 m (Fig. 8).

Average and standard deviation of slope gradient in the three geomorphic units (rock slopes, talus slopes, and the channel) throughout the TLS monitoring period were calculated from DEMs with various grid size. Regardless of the grid size, the average slope gradient of rock slopes and the channel were the highest and lowest among the three geomorphic units, respectively (Fig. 9a). As reported by previous studies (Loye et al., 2009), the average slope gradient of each geomorphic unit becomes gentler with an increase in the grid size, because small scale asperities of terrains (e.g., boulders) are smoothed when the grid size is larger (Fig. 9a). However, the relationship between grid size and slope gradient was slightly different among geomorphic units. For a grid size of ≥1.0 m, a histogram of the slope gradient in talus slopes was concentrated in a narrow range around the average value (i.e., >60% in the range between 34–40° for a grid size of 5.0 m; Fig. 9b). This slope angle is considered to represent $\alpha_1$, as reported in previous studies (Kirkby and Statham, 1975; Carson, 1977; Obanawa and Matsukura,

2008; Loye et al., 2009). The standard deviation of the slope gradient in the rock slope decreased with an increase in grid size when the grid size was smaller than the width of alternative strata in the study area (about 4.0 m; Imaizumi et al., 2015), and was almost constant when the grid size was larger than that. The standard deviation of the slope gradient in the channel also decreased with an increase in grid size when the grid size was smaller than 4.0 m, and was almost constant when the grid size was larger than 4.0 m. This inflection point of the trend (4.0 m) was the larger than the largest boulder size in the channel, changing with time from 1 to 2 m (Imaizumi et al., 2006; Imaizumi et al., 2016c).

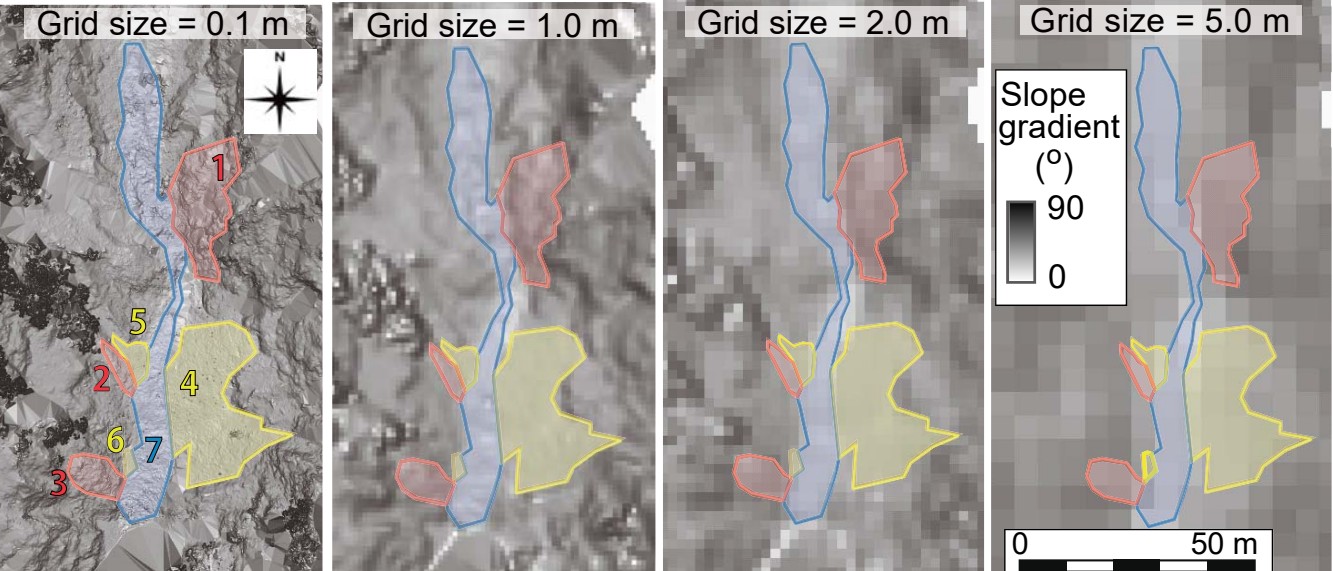

**Figure 8: Slope gradient map calculated after using terrestrial laser scanning (TLS) data on 14 May 2014, with various grid sizes. The location of the analysis area is shown in Fig. 2. Areas colored red, yellow, and blue indicate rock slopes, talus slopes, and a channel, respectively**

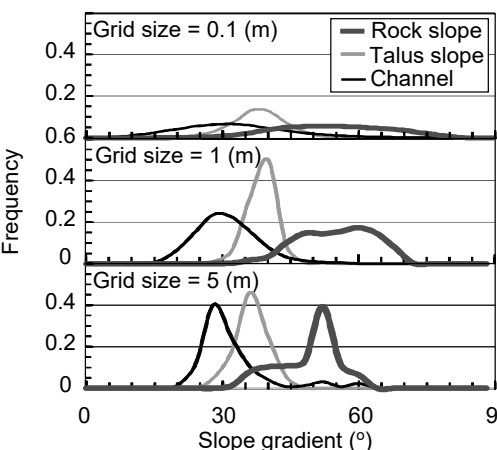

**Figure 9: Slope gradient of typical geomorphic units calculated from terrestrial laser scanning (TLS) derived digital elevation maps (DEMs) with various grid sizes. Topographic data in all measurement periods were used in the statistical analysis. (a) Changes in the average slope gradient of each geomorphic unit with an increase in the grid size. The slope gradient in all periods was averaged. The bars attached to the plots indicate the range of the standard deviation. (b) Histogram of the slope gradient calculated from TLS DEMs, with grid sizes of 0.1, 1.0, and 5.0 m. Integrals of the frequency for each geomorphic unit in Fig. 8b, which are categorized with 2 degree step, are all 1.**

## 5.3 Longitudinal channel topography

Longitudinal channel profiles obtained by the post-event surveys were analysed by GIS to find out the relationship between the debris flow type and channel gradient formed by the debris flow in the steep debris-flow initiation zone (Fig. 10). The channel topography between points A and E (total length of 100 m; Fig. 2) on 21 November 2012 was mainly formed by a debris flow on 30 September 2012, which was dominated by fully saturated flow. In contrast, the channel topography on 23 August 2015, was mainly formed by a debris flow on 6 August 2015, which was dominated by partly saturated flow. A small debris flow on 17 August 2015, which transported storage with a volume of <3,000 $m^3$, also affected a part of the channel topography measured on 23 August 2015; however, the affected area was limited (mostly a width of <5 m) because of its small discharge. Thus, we assumed that the channel profile in the survey section on 23 August 2015 reflected characteristics of the debris flow on 6 August 2015, rather than that on 17 August 2015.

When we compared channel bed profiles on 21 November 2012 and 23 August 2015, the channel bed level was clearly different (>3 m in depth) at the lowermost and uppermost reaches (within 25 m and from 75 to 90 m upstream of point A, respectively). In contrast, there was almost no change in the longitudinal channel profile (<1 m) in the middle reaches (within 30 m upstream of point C). Such differences in changes of the channel bed topography could also be identified in the cross-sectional profile. The cross-sectional profile at the lowermost and uppermost reaches (points B and D, respectively) was clearly different, while difference in the profile was not apparent at the middle reaches (point C; Figs. 10e, 10f, 10g). Channel

topography after many other debris flows in our monitoring site showed similar trend that the temporal changes in the channel bed topography in the lowermost and uppermost reaches was active, in contrast to the inactive riverbed changes in the middle reaches (Hayakawa et al., 2016).

The longitudinal profile of the channel gradient differed among debris flow events. The distribution of the channel gradient calculated for each 0.1 m section (the scale of a boulder) on November 21, 2012 (mainly formed by fully saturated flow), was clearly wider than that on 23 August 2015 (mainly formed by partly saturated flow; Fig. 10b, Table 3a). The p-value obtained by an F-test of the difference in the dispersion of the channel gradient between the two periods was <0.001. In contrast, the average slope gradient in the section between A and E was not significantly different between the two periods (23.1° and 25.4° for 21 November 2012, and 23 August 2015, respectively).

The difference in the amplitude of the channel gradient calculated for each 1.0 and 5.0 m section was also clear (Figs. 10c, 10d, Table 3a). The p-values obtained by an F-test of the difference in the dispersion of the channel gradient between the two periods was <0.001 and 0.065 for each 1.0 and 5.0 m section, respectively. Such temporal changes in the channel gradient was more evident in the sections with clear changes in the channel bed topography (from 0 to 25 and from 75 to 90 m from point A) compared to those without clear changes in the channel bed topography (from 25 to 75 and from 90 to 100 m from point A) (Tables 3b, 3c). Wide distribution of the slope gradient for each 5.0 m section on 21 November 2012 was mainly attributed to the step-pool topography formed by large boulders and the bedrock step-pool, while that for each 1.0 m section was attributed to such small-scale topographies together with the particle size of large boulders. The cross-sectional topography in the reaches without active channel bed deformation was narrowed by the massive and steep rock cliffs at the left bank (right side of the point C in Fig. 10f). High stream power attributed to the high flow depth in this section possibly restricted deposition of the sediment, resulting in inactive changes in the channel bed topography.

Periodic photography and field surveys after eight debris flows, which were successfully monitored by video cameras and TLCs, indicated that small-scale topography after debris flows mainly consisting of fully saturated flow are rugged compared to that after debris flows mainly consisting of partly saturated flow. This trend agrees with that observed by TLS surveys (Fig. 10, Table 3a).

The standard deviation of the channel gradient in the active channel bed deformation sections had a negative relationship with the volume of the storage (Fig. 11a). The p-values for the linear regressions with section lengths of 0.1, 1.0, and 5.0 m were 0.099, 0.026, and 0.021, respectively.

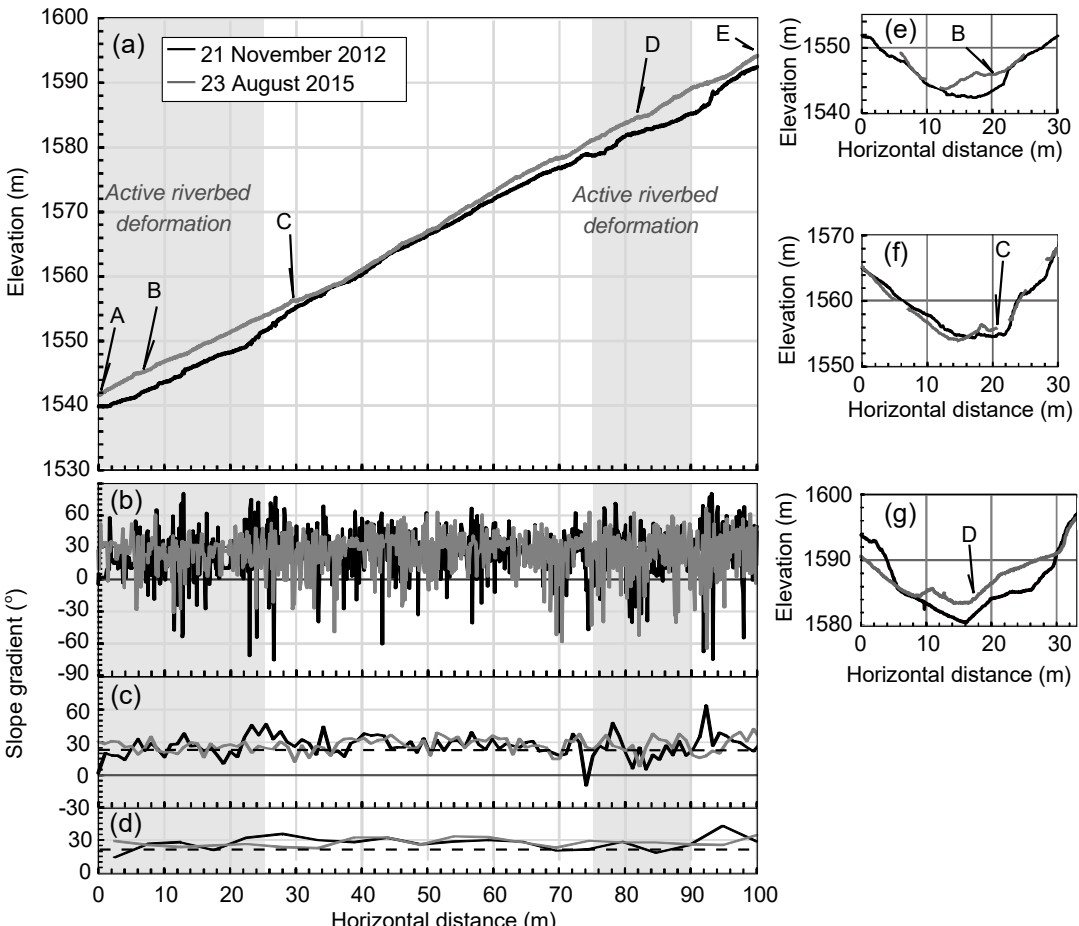

**Figure 10: Channel profiles and channel gradient mainly formed by fully saturated flow (measured on 21 November 2012 after the debris flow on 30 September 2012) and partly saturated flow (measured on 23 August 2015 after the debris flow on 6 August 2015), obtained from 0.1 m terrestrial laser scanning (TLS) derived digital elevation maps (DEMs). (a) Longitudinal channel profile in the section between A to E in Fig. 2. (b) Longitudinal changes in the channel gradient calculated for each section with a length of 0.1 m. (c) Longitudinal changes in the channel gradient calculated for each section with a length of 1.0 m. (d) Longitudinal changes in the channel gradient calculated for each section with a length of 5.0 m. (e) Cross-sectional profile of the channel at point B. (f) Cross-sectional profile of the channel at point C. (g) Cross-sectional profile of the channel at point D. Data at some cross-sections on 9 August 2015, are absent because of the failure of the TLS survey due to the weather conditions (panels e and f).**

**Table 3 Standard deviation of the channel gradient and proportional amount of channel sections in which the gradient was in the range between $\alpha_2$ and $\alpha_1$ throughout the entire channel section.**

(a) Entire channel section between points A to E

| Measurement date | Standard deviation of the channel gradient | | | Proportion of all sections between $\alpha_1$ and $\alpha_2$ | | |
|---|---|---|---|---|---|---|
| | 0.1 m interval | 1.0 m interval | 5.0 m interval | 0.1 m interval | 1.0 m interval | 5.0 m interval |
| 21 November 2012 | 23.7 | 10.3 | 5.8 | 0.31 | 0.52 | 0.80 |
| 23 August 2015 | 18.7 | 7.0 | 3.7 | 0.38 | 0.73 | 0.95 |

(b) Sections with active channel bed deformation (from 0 to 25 and from 75 to 90 m from point A).

| Measurement date | Standard deviation of the channel gradient | | | Proportion of all sections between $\alpha_1$ and $\alpha_2$ | | |
|---|---|---|---|---|---|---|
| | 0.1 m interval | 1.0 m interval | 5.0 m interval | 0.1 m interval | 1.0 m interval | 5.0 m interval |
| 21 November 2012 | 27.0 | 11.4 | 6.3 | 0.30 | 0.38 | 0.44 |
| 23 August 2015 | 17.9 | 7.2 | 3.4 | 0.48 | 0.71 | 0.89 |

(c) Sections without active channel bed deformation (from 25 to 75 and from 90 to 100 m from point A).

| Measurement date | Standard deviation of the channel gradient | | | Proportion of all sections between $\alpha_1$ and $\alpha_2$ | | |
|---|---|---|---|---|---|---|
| | 0.1 m interval | 1.0 m interval | 5.0 m interval | 0.1 m interval | 1.0 m interval | 5.0 m interval |
| 21 November 2012 | 20.9 | 8.6 | 5.1 | 0.34 | 0.61 | 0.90 |
| 23 August 2015 | 18.8 | 6.5 | 3.5 | 0.33 | 0.76 | 1.00 |

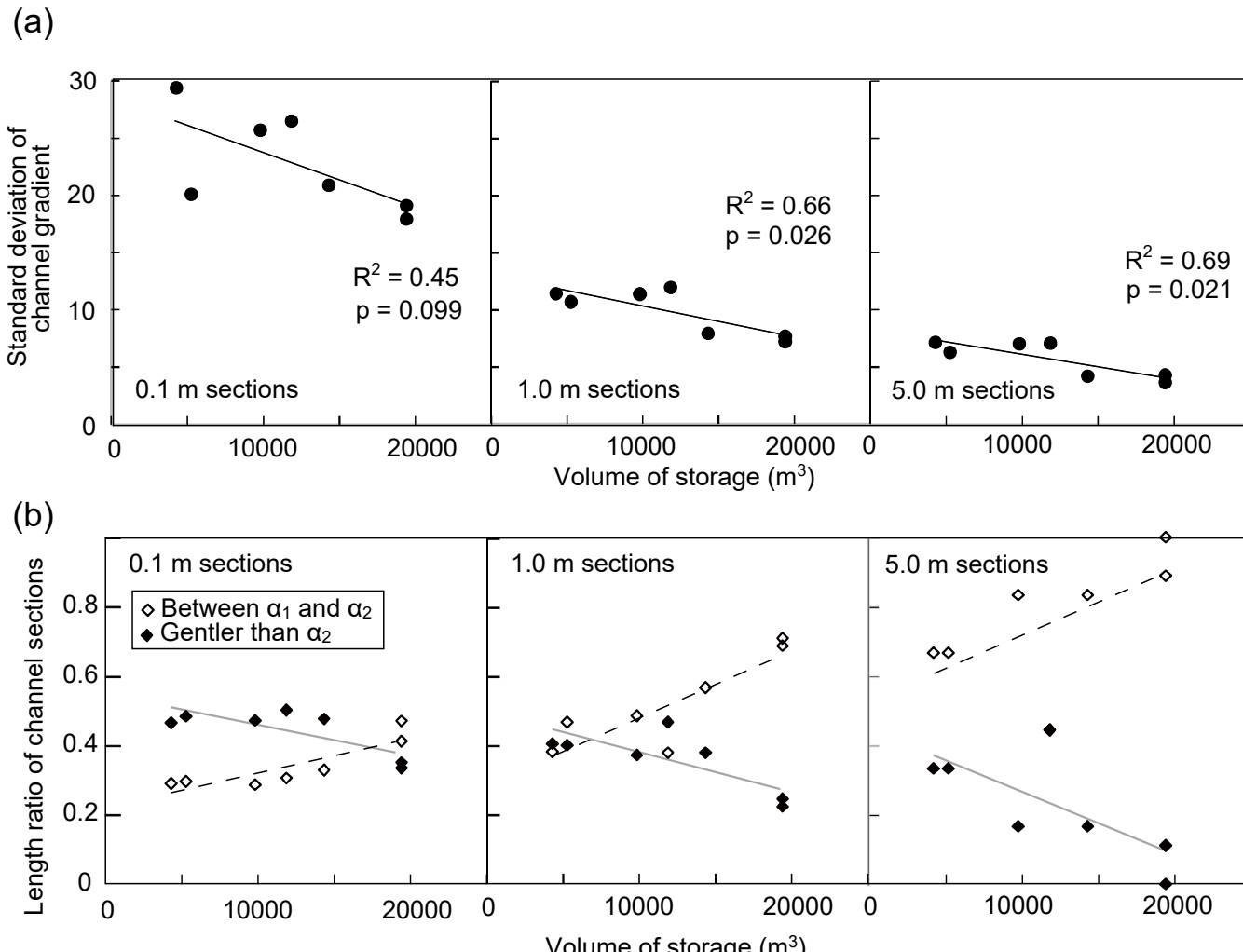

**Figure 11: Comparison between the volume of storage and channel topography. (a) Comparison of the volume of storage with the standard deviation of the channel gradient for each 0.1, 1.0, and 5.0 m section in the active channel bed deformation zones (from 0 to 25 m and from 75 to 90 m upstream of point A). (b) Comparison of the volume of storage with the length ratio of the channel sections, in which the channel gradient ranged between $\alpha_1$ and $\alpha_2$ (theoretical channel gradient for partly saturated debris flow) and gentler than $\alpha_2$ (theoretical channel gradient for fully saturated debris flow), in the active channel bed deformation zones. Topographic data before the first debris flow in each year were excluded from the analyses, because the channel gradient was possibly affected by the sediment supply from hillslopes, as well as the last debris flow of the previous year. The topography on August 17, 2014, was also excluded from the analysis because we failed to measure topography by terrestrial laser scanning (TLS) in the upper part of the active channel bed deformation zones because of fog. Linear regression lines for each relationship are also shown in the figure.**

As noted above, the theoretical channel gradient dividing partly and fully saturated debris flows ($\alpha_2$) can be obtained from Eq. (5). Because the dispersion of boulders in the partly saturated debris flows was not evident in video images, porosity (ratio of liquid and vapor phases, to be exact) in the partly saturated flow may not be higher than that in the saturated channel deposits. By assuming $\phi = 37.3°$ (average slope gradient of the talus slope for a grid size = 1.0 to 7.0 m), $n = 0.3$ (same as the porosity of channel deposits), $\gamma_s = 26000$, and $\gamma_w = 9800$, the $\alpha_2$ in the Ohya landslide was 22.2°. When we focused on the channel

sections with active channel bed deformation, the proportional amount of channel sections in which the gradient was in the range between $\alpha_2$ and $\alpha_1$ (theoretical channel gradient for partly saturated flow) in the entire channel section on 23 August 2015 (after mainly formed by partly saturated flow) was higher than on 21 November 2012 (after mainly formed by fully saturated flow), regardless of the interval of channel section for the analyses (Table 3b).

The proportional amount of all channel sections, in which the gradient was in the range between $\alpha_2$ and $\alpha_1$, throughout the entire channel section with clear changes in the channel bed topography was higher when the volume of storage was large (Fig. 11b). In contrast, the proportional amount of all channel sections, in which the channel gradient was gentler than $\alpha_2$, throughout the entire channel section decreased with an increase in the volume of sediment storage.

## 6. Discussion

**6.1 Factors controlling the debris flow type**

In contrast to debris-flow transportation zones in other torrents, in which partly saturated flow dominates just at the front of surges (McArdell et al., 2007; McCoy et al., 2010; Okano et al., 2012), our observations in the debris flow initiation zone of the upper Ichinosawa showed that overall debris flow surges were sometimes mainly composed of partly saturated flow (Table 2). Many sections in the upper Ichinosawa catchment are greater than 22.2° (Fig. 10), which is the theoretical channel gradient

needed for occurrence of a partly saturated debris flow in the Ichinosawa catchment. Although the $\alpha_2$ is different among torrents affected by soil parameters such as the internal angle of friction, debris flow initiation zones in other torrents, which are generally greater than 22.2° (e.g., VanDine 1985; McCoy et al., 2012), possibly satisfy the conditions for the occurrence of partly saturated flow.

The proportional duration of partly saturated flow in the overall surges had a positive relationship with the volume of storage

(Fig. 7). Thus, the volume of storage was a factor controlling not only the initiation of the debris flow, particularly in the supply limited basin (Bovis and Jakob, 1999; Jakob et al., 2005), but also the debris flow characteristics. The water content in the storage, which is considered an important factor controlling flow characteristics (Takahashi, 1991; Hürlimann et al., 2015), may affect such relationships. When a large volume of storage is present in a basin, a large amount of water is needed for saturation of the storage, which can be difficult to attain. In such cases, even if the sediment mass that initially moved in the

channel was saturated, storage along the channel eroded by the debris flow would not be fully saturated when the channel gradient $>\alpha_2$. Consequently, partly saturated flow can easily dominate when a large volume of storage accumulates in the channel. Such erosion of unsaturated sediment has also been reported in other channels (Berger et al., 2011b; McCoy et al., 2012). In contrast, storage can be easily saturated by smaller amounts of rainfall when there is only a small volume of storage in the basin. Thus, even when debris flows were not fully saturated in their initial stages, as they traveled downstream they

easily became saturated by the water supply from tributaries as well as the erosion of saturated deposits.

Our monitoring results also revealed that the proportional duration of partly saturated flow in a debris flow event was low during long-lasting rainfall events (e.g., typhoons and stationary fronts), while the duration was high during short-lasting

rainfall events (e.g., convective rainfall; Table 2, Fig. 5). The relationships between rainfall patterns and flow types have also been reported in other debris flow basins (Okano et al., 2012; Kean et al., 2013; Hürlimann et al., 2014). Okano et al. (2012) obtained a similar trend to that found in Ichinosawa catchment based on observations in the transportation zone. The front of a surge was completely saturated when rainfall intensity over a long duration (24 h) was high, while partly saturated flow occurred at the front of surges during short-term, intense rainfall events. Such a relationship between flow type and rainfall pattern can also be explained by the water content in the storage. During long-lasting rainfall events with high levels of cumulative rainfall, the amount of water in the storage can be high, resulting in a predominance of saturated flow. Water supply from tributaries during such long rainfall events also increases the water content in the debris flow (Okano et al., 2012). In contrast, the amount of water in the storage may be low during short-duration rainfall events with low levels of cumulative rainfall, resulting in the domination of partly saturated flow. Consequently, both amount of water supplied by rainfall and that of sediment storage need to be known when we consider water contents in the material affecting the debris flow type. Previous study reported changes in the flow characteristics during a debris flow event associated with the sediment supply from hillslopes (Staley et al., 2014; Zhou et al., 2015), possibly because of the increasing in the amount of sediment relative to the water in channels.

## 6.2 Influence of the sediment transport type on the slope gradient of terrains

The value of topographic indexes (e.g., slope gradient) are variable depending on the scale of the window size used in the GIS analyses, because factors affecting the indexes are different among the scales of topography (Schmidt and Andrew, 2005; Loye et al., 2009; Pirotti and Tarolli, 2010; Drăguţ and Eisank, 2011). In order to figure out appropriate window size for the analysis of the sediment transport type, we compared histograms of the slope gradient calculated with various window sizes each other (Fig. 9). As a result, the peak of the histogram of the slope gradient for each geomorphic unit was steep and high when the grid size was sufficiently large to eliminate the influence of small-scale roughness attributed to particle size (Fig. 9b). For example, histogram for the talus slope was steep when grid size was ≥1.0 m, which was larger than the maximum grain size of boulders on talus slopes (0.5 m). Histogram for the channel with grid size of 5.0 m was steeper and higher than that with grid size of 1.0 m, possibly affected by the size of the largest boulders in the channel, changing from 1 to 2 m associated with the sediment supply from hillslopes and sediment transport by debris flows (Imaizumi et al., 2006; Imaizumi et al., 2016c). Thus, when we analyse relationship between the type of sediment transport and the slope gradient of terrains, window size larger than the particle size, which can eliminate roughness attributed to the particle size, would be appropriate, because the particle size is considered to be not directly affected by the sediment transport type.

A large part of the histogram of the rock slopes was steeper than 40° for a grid size of >4.0 m (e.g., 80% of the slope-gradient histogram for a grid size of 5.0 m; Fig. 9b), indicating that fully unsaturated sediment transport, which occurs on slopes steeper than $\alpha_1$ (Eq. (4)), is dominant on rock slopes. In contrast, most of the channel was gentler than $\alpha_1$ for a grid size of >4.0 m (e.g., >85% of the slope-gradient histogram for a grid size of 5.0 m; Fig. 9b), indicating that fully and partly saturated debris flows are dominant in the channel. Such a trend was also apparent in the longitudinal profile of the channel (Fig. 10b). Based on the

statistics of the channel gradient for each 5.0 m channel section, the sum of the length ratio of the channel sections in the theoretical range of fully and partly saturated debris flows (between $\alpha_1$ and $\alpha_2$ ) in the active channel bed deformation zones was almost 1.0.

The longitudinal channel gradient changed with time and was affected by the predominant type of flow during the preceding debris flow events (Fig. 10). The proportional amount of channel sections between $\alpha_1$ and $\alpha_2$ (theoretical range of the channel gradient for partly saturated debris flow) among all active channel deformation sections was higher when a larger volume of storage was present in the catchment (Fig. 11b). Because partly saturated flow dominated when a large volume of storage was present in the catchment (Fig. 7), partly saturated debris flow likely formed channel topographies with a theoretical channel gradient for the partly saturated flow (Fig. 12). Similarly, the length of the channel sections in the theoretical range for fully saturated flow ($<\alpha_2$) was high when small amounts of storage accumulated in the channel (Fig. 11b), during periods when fully saturated flow dominated (Fig. 7). Thus, fully saturated flow tended to form channel sections gentler than $\alpha_2$ (Fig. 12). The standard deviation of the channel gradient after a debris flow mainly consisting of fully saturated flow was higher than that after a debris flow mainly consisting of partly saturated flow (Fig. 9, Table 3). In addition, the standard deviation of the channel gradient was high when small volumes of storage accumulated in the basin (Fig. 11a), implying that fully saturated flow tended to form a step-pool topography. Such trend was clear in the window size eliminating roughness attributed to the particle size of large boulders (i.e., 5.0 m sections). As a result of field surveys and periodic topography, step-pool topography formed by large boulders and the bedrock was generally seen after fully saturated flows. Since temporal changes in the channel gradient were not apparent in the larger-scale topography (i.e., channel gradient for entire sections with an active riverbed deformation with a length of 40 m), the decrease in the channel gradient due to fully saturated flow did not occur throughout the channel but was occasionally interrupted by steep sections formed by large boulders and bed rock, which cannot be eroded easily (Fig. 12).

Because we did not consider the dynamic mechanisms of the debris flow, including the collision of particles and the dynamic pressure of interstitial water (Coussot and Meunier, 1999; Takahashi, 2014), our analysis could not discuss unsteady nature of debris flows, such as temporal changes in the flow type during a debris flow event. In addition, temporary appearance of the unsaturated (or extremely dense) debris flows in gentler torrents does not agree with our analysis of static force (McArdell et al., 2007; McCoy et al., 2013; Okano et al., 2012). Depth profile of the dynamic force affected by the slope gradient and the particle size relative to the flow depth should be considered to complete explanation of such dense debris flows (Takahashi, 2007; Lanzoni et al., 2017). Nevertheless, our analysis results suggested that rough relationships between the predominant types of sediment transport and the slope gradient exist in the debris flow initiation zone.

| Volume of storage | Large | Small |
|---|---|---|
| Flow type | Mainly composed of partly saturated flow | Mainly composed of fully saturated flow |
| Channel topography | Bedrock | Bedrock |
| | $\alpha_2 < \alpha < \alpha_1$ | $\alpha < \alpha_2$  $\alpha < \alpha_2$  $\alpha < \alpha_2$ |

**Figure 12: Schematic diagram of the relationships between the volume of storage, flow type, and channel topography. $\alpha$ is the channel gradient, $\alpha_1$ is the theoretical boundary of the channel gradient between fully unsaturated and partly saturated sediment transport, and $\alpha_2$ is the theoretical boundary of the channel gradient between partly and fully saturated sediment transport.**

## 7 Summary and conclusion

We investigated the interaction between flow type (partly and fully saturated flows) and accumulation conditions of sediment storage (i.e., channel gradient and volume of storage) in the debris flow initiation zone at the Ohya landslide, Japan, using various methods, including a physical analysis of static force, field monitoring, periodical TLS surveys, and an estimation of the volume of storage using GIS. Our study revealed that both partly and fully saturated debris flows are important hydrogeomorphic processes in the initiation zone because of the steep terrain. The predominant type of flow is affected by the volume of storage as well as rainfall patterns, which control the amount of water in the storage. For example, fully saturated flow dominated when total volume of storage $< 10,000$ m$^3$, while partly saturated dominated when total of the storage $> 15,000$ m$^3$. In addition, small-scale channel gradients (on the order of meters) formed by erosion and deposition reflected the dominant flow type in the debris flow events. A large part of the channel sections after partly saturated debris flows were steeper than $22.2°$, while fully saturated debris flows tended to formed channel sections gentler than $22.2°$. Such relationship between flow type and channel gradient could be explained by a simple analysis of static force at the bottom of the sediment mass.

Our study implies that the volume of storage and the rainfall patterns, which control predominant debris flow type classified by the ratio of the depth of the saturated zone, are potential information to improve debris-flow warning system, since the flow characteristics (e.g., sediment concentration, velocity) is different among debris flow types. In addition, our study elucidated that the slope gradient of geomorphic units is the key factor in the estimation of predominant type of the sediment transport processes in the Ohya landslide, where stony debris flows occur due to the mobilization of storage laid on the steep channel. Therefore periodical measurement of the topography in debris flow initiation zones is considered to be essential for the better risk assessment of debris-flow hazards. Because the initiation mechanism of the debris flow and the important force inside of the flow (e.g., frictional force, grain collision) varies affected by the site conditions (e.g., grain size of material, slope gradient), monitoring data in other debris flow torrents are needed for the further understanding of interactions between the debris flow type and the accumulation conditions of storage.

**Acknowledgement**

This study was supported by JSPS Grant Numbers 25702014, 26292077, and 26282076. Airborne DEM was provided by Shizuoka River Office, Chubu Regional Bureau, Ministry of Land, Infrastructure, Transport and Tourism, Japan. We thank Hitoshi Saito for his comments improving structure of the manuscript. We are grateful to reviewers Carlo Gregoretti and

Michel Jaboyedoff who provided critical reviews that improved our paper. Editor Paolo Tarolli are thanked for his efforts on handling the manuscript.

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
