# Peer review of "Relationship between the accumulation of sediment storage and debris flow characteristics in a debris-flow initiation zone, Ohya landslide body, Japan"

_Natural Hazards and Earth System Sciences, 2017_

## Referee Comment (RC1) · C. Gregoretti (Referee) · 17 Feb 2017

**Overview**

The authors present a study that link the debris flow types to the morphological characteristics of the initiation zone and the entrainment and deposition processes to the debris flow types. The writer identified the following deficiencies:

1) The text, in some parts, is not fluid but hard to read. It looks like an ensemble of pieces more and less linked each other.

2) Authors through a static equilibrium-based relationship (equation 5), obtain angles corresponding to different debris flow types or the sediment concentration of debris flow corresponding to bed slope angles, if the other quantities are known. If used for dynamic computation (i.e. the sediment volume concentration of the flowing material) this equation is misleading. In the case of motion, the equation (5) is a bit different (Lanzoni et al., 2017) and is obtained through the ratio between the basal bed shear stress and the basal normal stress. Moreover in the case of flowing material, the angle $\phi$ is not the static friction angle but the quasi-static of dynamic friction angle (Lanzoni et al., 2017). Therefore, also the sentence "Thus, not only…..Eq(5)", is uncorrected. Therefore, I suggest the authors to write that an equation of the same structure of eq. (5) can be obtained through the ratio between the basal bed shear stress and the basal normal stress that have a similar structure to the ration between shear stress and strength (Lanzoni et al., 2017).

3) About triggering of debris flow, both the experiments of Gregoretti (2000) and the theoretical computations of Gregoretti (2008) clearly show that the entrainment of debris material into a water stream is provided by the surficial erosion of the debris layer rather than by the slide of the debris layer. Moreover, field works of Berti and Simoni (2005) , Cannon et al. (2008), Coe et al. (2008), Gregoretti and Dalla Fontana (2008), Theule et al. (2012), Hurliman et al. (2014), Degetto et al. (2015), Hu et al. (2016) point to a triggering mechanism where runoff erode the sediments and spread them along flow depth rather than the slide of a saturated mass. Recent works of Gregoretti et al. (2016) and Rengers et al. (2016) show that runoff descending from cliffs is an impulsive phenomenon characterized

by a peaked hydrograph and that the impact of peak against debris deposits entrain enough material to have a solid-liquid current. Finally, authors cited the work of Kean et al. (2013) and Navratil et al. (2013). In the first work, it is expressly written that debris flow initiation by surface runoff is different from the debris flow initiation by shallow landslide (the title of the work deals with runoff-generated debris flow), while in the second it is stated that debris flow is initiated by surface runoff rather than landslide. Therefore, if the authors intend to use their slide model, they should state that channelized debris flow initiate by runoff as a surficial erosion (cite the references above) and that they approximate it by a "slide" model using the calculation that Prancevic et al. (2014) show.

4) About 6.1, the writer agrees with authors that sediment availability determines the type of debris flows but has some concern about partially saturated debris flow even if stated by other authors (e.g. Iverson and Vallance, 2001). They could be very dense debris flows, where fluid phase is just under the surficial sediments. Measurements carried out at Illgraben (Mc Ardell et al., 2007) show a flow density of the front that approximates that of a saturated terrain. An alternative way is that of very dense debris flow rich of debris material and more fluid debris flow. About the outcome of the correspondence between short-lasting rainfall and partially saturated debris flows and that between long-lasting rainfalls and saturated debris flows, the field experiences in the initiation area Cancia debris flow given by Bernard et al. (2016) and Gregoretti et al. (2016) seem to contradict it. At Rovina di Cancia (Northeastern Italian Alps) an hyperconcentrated flow occurred about 11 days later a partially saturated debris flows (according to the author definition; see the video at the following link https://www.youtube.com/watch?v=oKQSZVwOuRo). The rainfall depths were just a bit smaller in the second event even if the terrain was not dry as for the first event (Gregoretti et al., 2016). The main reason is that after the first event, channel did not recharge and the quantity of entrained sediments in the second event in the initiation reach was at least an order smaller than that entrained during the first event.

The following are the detailed comments and specifications.

1. Page 1 – line 20: the sentence "The small-scale channel gradient…." is unclear.
2. Page 1 - line 26: has been
3. Page 1 - line 28: the word "activities" after "monitoring" is missing.

4. Page 2 - line 1: some parts of the sentence "Understanding ……….system" are unclear.

5. Page 2 – line 4: perhaps the reference Takahashi 2007 is better than Takahashi 1991.

6. Page 2 - line 18: add the reference Gregoretti and Dalla Fontana (2008).

7. Page 4 – line 5: what does it mean "is the exception ….. between α and ϕ?".

8. Page 4 – line 30: the sentence "The explanations…..2014)" is unclear as it regards the terrain formed by the sediment mass: what does it means? Moreover, the relationship given by Takahashi (2014) should be written.

9. Page 6 – lines 9-11: the sentence " Most of the channel…..low" is bad written and misleading.

10. Page 7 – line 8: P2 is not in fig. 2a but in fig. 2b, 142-143: the finding of Chen (1991) could be due to the increase of melting water, while in the present case debris flow has been triggered by rain storm.

11. Page 8 – lines 2 and 6: perhaps mean flow depth velocity or flow depth averaged velocity should be better than "mean velocity of all layers of the flow".

12. Page 9 – line 14: the sentence "The location of …" is bad written.

13. Page 11 – line 5 and following: which technique did used the authors to obtain the map of storage through the photographs taken at P1 and P4?

14. Page 12 – line 3: substitute "section between sites" with "reach between the sites".

15. Page 12 – line 9: "high value of total rainfall depth" instead of "high total rainfall"

16. Page 12 – line 10: "characterized by high intensity" instead of "of high intensity".

17. Page 12 – line 14: "varied between the events" instead of "differed between events".

18. Page 12 – lines 14-16: the sentence "For example……(Fig. 5)" should be rewritten as: For example in the event of 5 August 2008, 88% of debris flow surges (percentage respect to the total event duration) was composed of partly saturated flow, while in the event of 30 August 2004 (Fig. 5), 90% in time of the phenomenon was composed of fully saturated flow.

19. Page 12 – lines (16-19): sentence bad written: what does it mean the relation between camera location and debris flow characteristics?

20. Page 12 – lines (19-24): sentences very bad written; please rewrite them stating that the typology of rainfall determined the flow typology.

21. Page 14- lines (8-9): the writer does not understand the sentence "The roughness of the ground surface attributable to boulders……." How the roughness that is a length can be estimated by a slope gradient map? Moreover, the authors should introduce the definition of slope gradient or explain it.

22. Page 14 – line 13: authors should specify that obtained the DEM after using TLS data and the method they used for determining the slope gradient.

23. Page 14 – line 14: please place "the" before highest

24. Page 14 – line 17: talus slope is the slope gradient?

25. Page 15 – caption of Figure 7: calculated after using TLS data…..

26. Page 16 – lines (1-8): the writer does not understand what the authors mean. In other words, what does it the meaning of the channel topography forming after debris flow occurrence?. It means that all the channel was flooded by debris flow or that the debris flow changed the bed.

27. Page 16 – lines (9-14): even in these sentences the writer does not understand what the authors mean. Visual inspection of the two bed profiles of Figure 9a (are they from post-event surveys?) show that there is an high lowering of the bed profiles at the beginning and ending reaches. Authors should provide a much better description/comment of this figure.

28. Page 17 – caption of Figure 9: authors should specify that the profiles correspond to post-event surveys.

29. Pages 19-20. All the comments about $\alpha_1$, $\alpha_2$ the storage volume and slope gradient should consider that the middle part of the channel was not interested by erosion phenomena. Therefore, the authors should explain a reason and then exclude it from the further comments. Moreover, the sentence between pages 19 and 20 is not clear.

30. Page 21 - lines 14-15. The explanation on the effect of grid size on hystogram shape should be coupled with some estimation of roughness (i.e. median grain size or something else) . Moreover, the authors should initiate the subsection explaining the reasons they produced the histograms.

31. Pages 21-22: the writer does not understand well the scope of the written sentences: authors should clearly rewrite them. For istance, fully saturated debris flow removed only fine sediments while partially saturated debris flows washed out entire bottom reaches. The writer thinks that the amount of entrained material depend also on the main characteristics of the of the solid-liquid current: flow depth, sediment concentration and velocity that in the case of coarse grained debris flow depends also on the runoff discharge (Lanzoni et al., 2017).

Bernard M., Stancanelli L., Berti M., Simoni A., Gregoretti C., Lanzoni S. (2016) *Field results from the runoff generated debris flows occurred at Rovina di Cancia (Venetian Dolomites)* XXXV Convegno di Idraulica e Costruzioni Idrauliche – Bologna.

Berti, M., and A. Simoni (2005), Experimental evidences and numerical modelling of debris flow initiated by channel runoff, *Landslide*, 2, 171--182.

Cannon, S., Gartner J.E., Wilson, R.C., Bowers, J.C., Laber, J.L. 2008. Storm rainfall conditions for floods and debris flows from recently burned areas in Southwestern Colorado and Southern California. *Geomorphology*, 96, 250-269.

Coe, J.A., Kinner D.A., Godt, J.W., 2008. Initiation conditions for debris flows generated by runoff at Chalk Cliffs, central Colorado. *Geomorphology*, 96, 270-297.

Degetto, M., Gregoretti, C., and Bernard M. 2015. Comparative analysis of the differences between using LiDAR contour-based DEMs for hydrological modeling of runoff generating debris flows in the Dolomites. *Frontier in Earth Sciences*. 3:21, doi:10.3389/feart.2015.00021

Gregoretti, C., Dalla Fontana G., 2008. The triggering of debris flow due to channel-bed failure debris flow in some alpine headwater basins of the Dolomites: analyses of critical runoff. *Hydrological Processes*. 22, 2248-2263.

Gregoretti C., Degetto M., Bernard M., Crucil, G., Pimazzoni A., De Vido G., Berti M., Simoni A. Lanzoni S. Runoff of small rocky headwater catchments: Field observations and hydrological modeling. *Water Resources Research*. 52(8) doi: 10.1002/2016WR018675

Hu, W., Dong, X.J., Wang, G.H., van Asch T.W.J. and Hicher P.Y. 2016. Initiation processes for run-off generated debris flows in the Wenchuan earthquake area of China, *Geomorphology*. 253, 468-477. http://dx.doi.org/10.1016/j.geomorph.2015.10.024

Hurlimann M., Abanco C., Moya, J., Vilajosana I. (2014). Results and experiences gathered at the Rebaixader debris-flow monitoring site, Central Pyrenees, Spain. *Landslides*. doi:10.1007/s10346-013-0452-y 161-175

Kean J.W., Staley D.M., Leeper R.J., Schmidt K.M., Gartner J.E. (2012). A low-cost method to measure the timing of postfire flash floods and debris flows relative to rainfall. *Water Resources Research*, 48, W05516, doi:10.1029/2011WR011460

Lanzoni S., Gregoretti C., Stancanelli L. (2017) Coarse-grained debris flow dynamics on erodible beds. *Journal of Geophysical Research: Earth Surface* doi*: 10.1002/2016JF004046

Rengers, F.K., L.A. McGuire, J.W. Kean and D.E. Hobley (2016), Model simnulations of flood and debris flow timing in steep cachments after wildfire, *Water Resources Research*, 52, doi:10.1029/2015WR018176.

Takahashi, T. (2007), Debris Flow: Mechanics, Prediction and Countermeasures, Balkema-proceedings and monographs in engineering, water and earth sciences, Taylor \& Francis.

Theule, J.I., Liebault, F., Loye, A., Laigle, D., and Jaboyedoff, M., 2012. Sediment budget monitoring of debris flow and bedload transport in the Manival Torrent, SE France. *Natural Hazard Earth Sciences*, 12, 731-749.

---

## Referee Comment (RC2) · C. Gregoretti (Referee) · 10 Mar 2017

dear author

after the editor decision, I will read the revised paper.

About your replay:

"We checked Gregoretti et al. (2016, WRR) but could not find statements on the flow characteristics and rainfall pattern triggering debris flow."

See the last sentences of page 8154, Table 4 and Figure 10 (panels c and d)

[Figure]

best wishes

Carlo Gregoretti
* * *

---

## Author Comment (AC1) · 10 Mar 2017

**Reply for review comments**

We sincerely thank you for the efforts you have made to improve our submission to *Natural Hazards and Earth System Sciences*. We have responded to all review comments and have made appropriate modifications to our manuscript related to these comments as detailed in the following paragraphs. The blue-highlighted sentences are the review comments; sentences in black represent our responses to these review comments.

The authors present a study that link the debris flow types to the morphological characteristics of the initiation zone and the entrainment and deposition processes to the debris flow types. The writer identified the following deficiencies:

1) The text, in some parts, is not fluid but hard to read. It looks like an ensemble of pieces more and less linked each other.

We have checked entire manuscript again, and revised sections, which are not linked each other (e.g., explanation about the static model from pg. 4 line 8 to pg. 5 line 20). In addition, we have made a substantial revision in section 6.2, because the scope of the discussion was not clear in the previous version of the manuscript.

2) Authors through a static equilibrium-based relationship (equation 5), obtain angles corresponding to different debris flow types or the sediment concentration of debris flow corresponding to bed slope angles, if the other quantities are known. If used for dynamic computation (i.e. the sediment volume concentration of the flowing material) this equation is misleading. In the case of motion, the equation (5) is a bit different (Lanzoni et al., 2017) and is obtained through the ratio between the basal bed shear stress and the basal normal stress. Moreover in the case of flowing material, the angle ϕ is not the static friction angle but the quasi-static of dynamic friction angle (Lanzoni et al., 2017). Therefore, also the sentence "Thus, not only…..Eq(5)", is uncorrected. Therefore, I suggest the authors to write that an equation of the same structure of eq. (5) can be obtained through the ratio between the basal bed shear stress and the basal normal stress that have a similar structure to the ration between shear stress and strength (Lanzoni et al., 2017).

Some governing equations for the debris flow in previous studies explained that only static frictional force (coulomb force), which can be obtained by the volume concentration of the flowing material, acts on the surface of the erosible bed (e.g., Egashira et al., 2001; Takahashi, 2014). However, as the reviewer pointed out, other studies consider flow dynamics differently. In any of these cases, we think the ratio between the shear stress and the normal stress at river bed is an important factor controlling flow characteristics. We have added the sentence suggested by the reviewer (pg. 4, lines 23-25). The sentence in p. 4 line 22-23 intended to mention that other types of sediment transport processes also satisfy the Eq. (5). We have revised the sentence to clarify what we mean (pg. 4, lines 26-27).

3) About triggering of debris flow, both the experiments of Gregoretti (2000) and the theoretical computations of Gregoretti (2008) clearly show that the entrainment of debris material into a water stream is provided by the surficial erosion of the debris layer rather than by the slide of the debris layer. Moreover, field works of Berti and Simoni (2005) , Cannon et al. (2008), Coe et al. (2008), Gregoretti and Dalla Fontana (2008), Theule et al. (2012), Hurliman et al. (2014), Degetto et al. (2015), Hu et al. (2016) point to a triggering mechanism where runoff erode the sediments and spread them along flow depth rather than the slide of a saturated mass. Recent works of Gregoretti et al. (2016) and Rengers et al. (2016) show that runoff descending from cliffs is an impulsive phenomenon characterized by a peaked hydrograph and that the impact of peak against debris deposits entrain enough material to have a solid-liquid current. Finally, authors cited the work of Kean et al. (2013) and Navratil et al. (2013). In the first work, it is expressly written that debris flow initiation by surface runoff is different from the debris flow initiation by shallow landslide (the title of the work deals with runoff-generated debris flow), while in the second it is stated that debris flow is initiated by surface runoff rather than landslide. Therefore, if the authors intend to use their slide model, they should state that channelized debris flow initiate by runoff as a surficial erosion (cite the references above) and that they approximate it by a "slide" model using the calculation that Prancevic et al. (2014) show.

Many debris flows in the Ohya landslide were also initiated by the erosion of the debris material by the surface runoff (Imaizumi et al., 2016). As the reviewer points out, erosion by the surface runoff is the predominant initiation mechanism of the debris flow. In the Ohya landslide, sliding of the debris material was also monitored. Our monitoring site is possibly in the traditional slope gradient range from fluvial processes to the failure of the debris deposits when we consider initiation mechanism of the debris flow like Prancevic et al. (2014). As the review suggested, we have added statements that many debris flows in the Ohya landslide and other debris flow torrents were triggered by the surface erosion. In addition, we noted that relationships between topography and type of sediment transport were discussed by approximation of the sediment transport type by the simple static models (pg. 5, lines 13-20).

Imaizumi, F., Tsuchiya, S. & Ohsaka, O. (2016) Field observations of debris-flow initiation processes on sediment deposits

in a previous deep-seated landslide site, Journal of Mountain science, 13: 213. doi:10.1007/s11629-015-3345-9

About 6.1, the writer agrees with authors that sediment availability determines the type of debris flows but has some concern about partially saturated debris flow even if stated by other authors (e.g. Iverson and Vallance, 2001). They could be very dense debris flows, where fluid phase is just under the surficial sediments. Measurements carried out at Illgraben (Mc Ardell et al., 2007) show a flow density of the front that approximates that of a saturated terrain. An alternative way is that of very dense debris flow rich of debris material and more fluid debris flow. About the outcome of the correspondence between short-lasting rainfall and partially saturated debris flows and that between long-lasting rainfalls and saturated debris flows, the field experiences in the initiation area Cancia debris flow given by Bernard et al. (2016) and Gregoretti et al. (2016) seem to contradict it. At Rovina di Cancia (Northeastern Italian Alps) an hyperconcentrated flow occurred about 11 days later a partially saturated debris flows (according to the author definition; see the video at the following link https://www.youtube.com/watch?v=oKQSZVwOuRo). The rainfall depths were just a bit smaller in the second event even if the terrain was not dry as for the first event (Gregoretti et al., 2016). The main reason is that after the first event, channel did not recharge and the quantity of entrained sediments in the second event in the initiation reach was at least an order smaller than that entrained during the first event.

Thank you for telling us an interesting video in YouTube. As the reviewer comments, density of debris flow can be very high (close to that of saturated terrain) particularly at the head of surges even in the gentler channel (McArdell et al., 2007; McCoy et al., 2010; Okano et al., 2012). Unfortunately, we do not have data related to the internal structure of the debris flow (e.g., thickness of the unsaturated layer, depth profiles of solid concentration and flow velocity). Thus, we cannot deeply discuss dynamic mechanism of the flow. We have revised statements on the partly saturated debris flow in section 6.1 to clarify relationship between results in our study and previous studies (pg.20, lines 11-18). We have also discussed limitation in our analyses in the end of discussion (pg. 22, lines 21-27). As presented in Takahashi (2007) and Lanzoni et al. (2017), we think depth profile of the dynamic force affected by the slope gradient and the particle size relative to the flow depth should be considered to complete explanation of the dense debris flows. On the other hand, our simple static-force analyses and results of the field monitoring implied that unsaturated sediment transport frequently occur in the steep terrains. We think that point is worth to be emphasized in this paper.

We checked Gregoretti et al. (2016, WRR) but could not find statements on the flow characteristics and rainfall pattern triggering debris flow. However, we agree with the reviewer's comment that recharge of the sediments into valley bottom affect volume of the sediment entrained by debris flows. Our monitoring results also implies that volume of debris flow material affects flow characteristics (Fig. 6). In the end of the section 6.1, we have emphasized importance of the amount of the debris flow material and that of water supplied on the debris flow type (pg. 21, lines 10-14). Grain size, amount of debris flow material, and topography may affect difference in the flow characteristics and initiation mechanisms among debris flow torrents. In the end of the paper (in conclusion), we have added statements that we need to collect data in other debris flow torrents with different site conditions (e.g., grain size, slope gradient) for the further understanding of interactions between the debris flow characteristics and the topography (pg. 23, lines 20-24).

1. Page 1 – line 20: the sentence "The small-scale channel gradient…." is unclear.
We have revised the sentence. Statements on characteristics of channel morphology (slope gradient) were added (pg. 1, lines 23-25).

2. Page 1 - line 26: has been
We have replaced "have" to "has" (pg. 1, line 28).

3. Page 1 - line 28: the word "activities" after "monitoring" is missing.
As suggested by the reviewer, we have added "activities" (pg. 2, line 1).

4. Page 2 - line 1: some parts of the sentence "Understanding ………..system" are unclear.
We have revised the sentence. Now the sentence is "factors affecting debris-flow characteristics in the initiation zone (e.g., temporal changes in the solid fraction) is still unclear" (pg. 2, lines 3-4).

5. Page 2 – line 4: perhaps the reference Takahashi 2007 is better than Takahashi 1991.
We have replaced the reference "Takahashi, 1991" to "Takahashi, 2007" (pg. 2, line 6).

6. Page 2 - line 18: add the reference Gregoretti and Dalla Fontana (2008).
We have added the reference Gregoretti and Dalla Fontana (2008) (pg. 2, line 23).

7. Page 4 – line 5: what does it mean "is the exception ….. between α and φ?".
We have revised the explanation about pyroclastic flow. The relationship between α and φ expressed as Eq. (4) cannot be applicable to the pyroclastic flow (pg. 4, lines 8-13).

8. Page 4 – line 30: the sentence "The explanations…..2014)" is unclear as it regards the terrain formed by the sediment mass:

what does it means? Moreover, the relationship given by Takahashi (2014) should be written.
The sentence explained relationship between solid fraction and channel gradient. In order to clarify the relationship between solid fraction and channel gradient, we have added Eq. (6). In addition, we replaced "terrain" to the "channel gradient" (pg. 5, lines 5-6).

9. Page 6 – lines 9-11: the sentence " Most of the channel…..low" is bad written and misleading.
This sentence explains the channel bed condition in the monitoring site. We have improved the sentence (pg. 7 line 2-4).

10. Page 7 – line 8: P2 is not in fig. 2a but in fig. 2b
We have removed "a" after "Fig. 2" (pg. 8, line 7).

11. Page 8 – lines 2 and 6: perhaps mean flow depth velocity or flow depth averaged velocity should be better than "mean velocity of all layers of the flow".
We have replaced "mean velocity of all layers of the flow" to "flow depth averaged velocity" (pg. 8, line 14).

12. Page 9 – line 14: the sentence "The location of …" is bad written.
We have revised the sentence. Now the sentence is "The location of this scan position varied with time because it was in the valley bottom where the topography changed due to erosion and deposition by debris flows."(pg. 10, lines 4-5).

13. Page 11 – line 5 and following: which technique did used the authors to obtain the map of storage through the photographs taken at P1 and P4?
We have added detailed explanation about the estimation method of the storage (pg. 11, line 14 - pg. 12, line 1).

14. Page 12 – line 3: substitute "section between sites" with "reach between the sites".
We have replaced "section between sites" to "reach between the sites" (pg. 12, line 7).

15. Page 12 – line 9: "high value of total rainfall depth" instead of "high total rainfall"
We have replaced "high total rainfall" to "high value of total rainfall depth" (pg. 12, line 13).

16. Page 12 – line 10: "characterized by high intensity" instead of "of high intensity"
We have replaced "of high intensity" to "characterized by high intensity" (pg. 12, line 14).

17. Page 12 – line 14: "varied between the events" instead of "differed between events".
We have replaced "differed between events" to "varied between the events" (pg. 12, line 18).

18. Page 12 – lines 14-16: the sentence "For example……(Fig. 5)" should be rewritten as: For example in the event of 5 August 2008, 88% of debris flow surges (percentage respect to the total event duration) was composed of partly saturated flow, while in the event of 30 August 2004 (Fig. 5), 90% in time of the phenomenon was composed of fully saturated flow.
We have rewritten the sentence as suggested by the reviewer (pg. 12, lines 18-20).

19. Page 12 – lines (16-19): sentence bad written: what does it mean the relation between camera location and debris flow characteristics?
Originally we meant that difference in the channel gradient among camera locations potentially affects difference in the flow type. However, the channel gradient changes with time even at the same camera location. Therefore, now we think first half of the sentence was not needed (pg. 12, line 20).

20. Page 12 – lines (19-24): sentences very bad written; please rewrite them stating that the typology of rainfall determined the flow typology.
We have rewritten the sentence. We think the sentence is easy to read now (pg.12, lines 22-25).

21. Page 14- lines (8-9): the writer does not understand the sentence "The roughness of the ground surface attributable to boulders……." How the roughness that is a length can be estimated by a slope gradient map? Moreover, the authors should introduce the definition of slope gradient or explain it.
We think the term "roughness" was not appropriate. We have revised the sentence. We also explained how the slope gradient was calculated (pg. 14, line 8-10).

22. Page 14 – line 13: authors should specify that obtained the DEM after using TLS data and the method they used for determining the slope gradient.
We have added explanations how to obtained DEMs from the TLS point clouds (pg. 10, line 19-21). We have also presented calculation method of the slope gradient (pg. 14, lines 8-9). We used the method proposed by Horn (1981).

23. Page 14 – line 14: please place "the" before highest

We have added "the" in front of the "highest" (pg. 14, line 15).

24. Page 14 – line 17: talus slope is the slope gradient?
We have rephrased the sentence. Now the phase is "a histogram of the slope gradient in talus slopes…" (pg. 14, line 18).

25. Page 15 – caption of Figure 7: calculated after using TLS data…..
We have revised the figure caption as suggested by the reviewer (pg. 15)

26. Page 16 – lines (1-8): the writer does not understand what the authors mean. In other words, what does it the meaning of the channel topography forming after debris flow occurrence?. It means that all the channel was flooded by debris flow or that the debris flow changed the bed.
We have added explanation about the aim of analyses on the longitudinal channel topography (pg. 16, lines 2-4). We tried to find out relationships between the flow type and the longitudinal profile of the channel (channel gradient) formed by the debris flow in the steep debris-flow initiation zone. Most part of the channel topography in the section between points A and E was changed by the debris flows on 30 September 2012 and 6 August 2015. In contrast, channel topography was not largely changed by the small debris flow on 17 August 2015. Therefore, we assumed that the channel profile on 23 August 2015 reflected characteristics of the debris flow on 6 August 2015, rather than that on 17 August 2015 (pg. 16, lines 9-10).

27. Page 16 – lines (9-14): even in these sentences the writer does not understand what the authors mean. Visual inspection of the two bed profiles of Figure 9a (are they from post-event surveys?) show that there is an high lowering of the bed profiles at the beginning and ending reaches. Authors should provide a much better description/comment of this figure.
We intended to explain similar characteristics as the reviewer comments. We have rewritten the paragraph (pg. 16, lines 11-18). The topography was obtained by the post-event surveys (pg. 16, line 2).

28. Page 17 – caption of Figure 9: authors should specify that the profiles correspond to post-event surveys.
We have added timing of the debris flow events in the figure caption (pg. 17). Thus, now it is clear that the profile was measured after the debris flow events.

29. Pages 19-20. All the comments about α1, α2 the storage volume and slope gradient should consider that the middle part of the channel was not interested by erosion phenomena. Therefore, the authors should explain a reason and then exclude it from the further comments. Moreover, the sentence between pages 19 and 20 is not clear.
We have discussed the reason why the channel bed deformation was not clear in the middle part of the profile (pg. 16, line 33-pg. 17, line 2). Channel was narrowed because of the massive rock cliffs at the left bank. Therefore, flow depth in this section may exceeds that in the other sections. High stream power attributed to the high flow depth may restricted deposition of the sediments, possibly resulting in the small channel bed deformation. We also excluded a sentence about inactive section (pg. 20, line 2). We revised the sentence which, was between pages 19 and 20 (pg. 19, line 14-pg. 20, line 2).

30. Page 21 - lines 14-15. The explanation on the effect of grid size on hystogram shape should be coupled with some estimation of roughness (i.e. median grain size or something else) . Moreover, the authors should initiate the subsection explaining the reasons they produced the histograms.
We have added discussion between the shape of the histograms and grain size (pg. 21, lines 20-25). We think that the influence of the grain size on the slope gradient should be eliminated when we discuss relationship between the slope gradient and the sediment transport type (pg, 21, lines 25-28). Analyses of the histogram provides us idea on the influence of roughness attributed to the grain size on the slope gradient. We have added explanation about importance such analyses in the top of the section (pg. 21, lines 16-20).

31. Pages 21-22: the writer does not understand well the scope of the written sentences: authors should clearly rewrite them. For istance, fully saturated debris flow removed only fine sediments while partially saturated debris flows washed out entire bottom reaches. The writer thinks that the amount of entrained material depend also on the main characteristics of the of the solid-liquid current: flow depth, sediment concentration and velocity that in the case of coarse grained debris flow depends also on the runoff discharge (Lanzoni et al., 2017).
We agree with the reviewer's comment that the amount of entrained material depends on the flow depth, sediment concentration and velocity (this can be expressed as a function of flow depth and sediment concentration). In this section, we did not discuss amount of entrained material, but we discussed relationship between the (channel) topography and the type of the sediment transport. In order to make the topic clear, we have changed title of the section to "Influence of the sediment transport type on the slope gradient of terrains". The topic about the selective transport of the channel deposits would be not necessary in this paper. In addition, that topic may have obscured scope of the discussion. Therefore, we have removed the statements on the selective transport. We also revised the end of the section to emphasize importance of the relationships between the channel topography and type of sediment transport (pg. 22, lines 28-29).

**Interactions between the accumulation of sediment storage and debris flow characteristics in a debris-flow initiation zone, Ohya landslide body, Japan**

Fumitoshi Imaizumi[1], Yuichi S. Hayakawa[2], Norifumi Hotta[3], Haruka Tsunetaka[4], Okihiro Ohsaka[1], Satoshi Tsuchiya[1]

[1]Faculty of Agriculture, Shizuoka University, Shizuoka, 422-8529, Japan
[2]Centre for Spatial Information Science, The University of Tokyo, Kashiwa, 277-0871, Japan
[3]Faculty of Life and Environmental sciences, University of Tsukuba, Tsukuba, 305-8572, Japan
[4]Graduate School of Life and Environmental sciences, University of Tsukuba, Tsukuba, 305-8572, Japan

*Correspondence to*: Fumitoshi Imaizumi (imaizumi@shizuoka.ac.jp)

**Abstract.** Debris flows often occur in steep mountain channels, and can be extremely hazardous as a result of their destructive power, long travel distance, and high velocity. However, their characteristics in the initiation zones, which could possibly be affected by temporal changes in the channel topography associated with sediment supply from hillslopes and the evacuation of sediment by debris flows, are poorly understood. Thus, we studied the interaction between the flow characteristics and the topography in an initiation zone of debris flow at the Ohya landslide body in Japan using a variety of methods, including a physical analysis, a periodical terrestrial laser scanning (TLS) survey, and field monitoring. Our study clarified that both partly and fully saturated debris flows are important hydrogeomorphic processes in the initiation zones of debris flow because of the steep terrain. The predominant type of flow varied temporally and was affected by the volume of storage and rainfall patterns. The small-scale channel gradient (on the order of meters) formed by debris flows differed between the predominant flow types during debris flow events. The relationship between flow type and the slope gradient could be explained by a simple analysis of the static force at the bottom of the sediment mass. Partly saturated debris flow formed channel topographies with a theoretical channel gradient for the partly saturated flow (22.2 – 37.3° in the Ohya landslide), while fully saturated debris flow formed channel topographies with gentler channel gradient (<22.2° in the Ohya landslide).

[revised manuscript text omitted]

---

## Referee Comment (RC3) · C. Gregoretti (Referee) · 13 Mar 2017

dear authors

the phenomenon type (debris flow and hyperconcentrated flows) refers to what occurred in the monitored initiation area. The hyperconcentrated flow evolved in debris flow about 100 m downstream because of the progressive entrainment of bed material. About rainfall, in the second event (4th of August) terrain was in the intermediate saturated condition AMCII while in the first event it was AMCI (see Table 4 of Gregoretti et al., 2016). In fact runoff peak and volume of the second event are larger. The total

mobilized sediment volume of the second event is about 10% smaller than that mobilized during the first event even if runoff peak and volume are smaller. This is explained by the different sediments availability, because the second event occurred before the channel recharged.

best wishes

Carlo Gregoretti

---

## Short Comment (SC1) · 13 Mar 2017

Dear Reviewer

Thank you for telling us location of the statements about the two rainfall events in Cancia. As the reviewer points out, rainfall depth of the event on 4 August 2015, which triggered hyperconcentrated flow, was a little smaller than that of the event on 23 Jul 2015, which triggered debris flow. Small amount of sediment storage on 4 August 2015 may have affected appearance of the hyperconcentrated flow. As we emphasized in the revised manuscript, both amount of rainfall and that of debris-flow material in the

channel should be considered to explain difference in the flow type. Analyses of debris flows in other torrents would be needed for the further understanding of the factors controlling the flow type.

Sincerely yours,

Fumitoshi Imaizumi

---

## Short Comment (SC2) · 14 Mar 2017

Dear Reviewer

Thank you for detailed explanation on characteristics of two debris flow events. Our monitoring at multiple sites also revealed that flow characteristics in the upper and lower reaches are sometimes different because of the erosion and deposition of the sediment during downstream migration. We have not published the result as a paper, but we would like to publish in the near future. Moisture conditions of deposits antecedent to rainfall would be another factor affecting initiation of debris flows and flow

characteristics. However, initiation condition of debris flows and flow characteristics in the Ohya landslide can be explained by the characteristics of the rainfall triggering debris flow (this manuscript; Imaizumi et al., 2005, Canadian Geotechnical Journal). Influence of antecedent rainfall (moisture) seems to be not large. Characteristics of debris flow material, climate, and regional rainfall pattern possibly affect such small influence of the antecedent moisture conditions.

Sincerely yours,

Fumitoshi Imaizumi

---

## Referee Comment (RC4) · M. Jaboyedoff (Referee) · 31 Mar 2017

Dear Editor,

Please find here below my review of the paper nhess-2017-20:

This study is an analysis is an attempt to characterize better the source zones of debris-flow in relationship with the storage slope angle and precipitations. The data were acquired in a catchment, which includes a large landslide. The monitoring was performed using TLS, video cameras, rainfall gauges and pressure sensor. In addition, field works were performed. The paper tries link saturation of the sediment, the volume

[Figure]

of the initiation zones, the type of flow and slope angles.

In conclusion, the flow characteristics be explained by the interplay of rainfall patterns saturating or not large or small volume of sediment. The slope gradient can be linked with the above conditions by the Takashi formulas. It shows that the fully saturated debris-flows create low gradient profile with breaks, while the partially saturated ones create constant steeper gradient.

General comments

The paper presents interesting results, but sometimes it is rather difficult to follow. It was difficult for me to write a summary, maybe the findings are not enough underlined and strengthened. In my opinion the authors have to read there paper carefully again simultaneously with a colleague that is not involved in the paper in order to clarifies make the text easier to read.

The abstract is as well no very informative.

The figures about the site are often too small and too dark.

Information about rainfall are lacking such as IDF or other information. In addition, the relationship of the debris flows with the landslide is not really described.

In my opinion if the author clarify the text and make more easy to read this will be valuable paper about debris-flow behaviours.

Specific comments

P2 line 1: "debris" instead of "decries"

P2 line 4: "Hungr" instead of "Hunger"

P4 line 14: this must be explained.

P4 lines 18-19: more explanations about dispersion

Fig 1 caption: remind the letters meanings.

[Figure]

P6 line 10-14: to introduce this subject reference to Theule paper in NHESS can be introduced in the introductive section.

Figure 2: limit of the landslide are missing or unclear.

Page 7 line 13- p8: line 4: more explanations are needed to explain how these parameters are evaluated.

P8: this page is not well structured difficult to follow. For instance the ultrasonic sensor are used to measure the surface height are explained at the end but already introduced at the beginning of the page.

P8: line 23: where are installed the pressure sensor.

P9 lines 9-10: this accuracy was checked or it is it the manufacturer data?

P10 line 4: only two target were used ???? not 3 minimum?

P10 first lines are repeated.

Table 1: please add the point spacing of the cloud points.

P10 line 10: original density of points is needed here

P10 lines 21-23: unclear please clarify.

P11 line 5: how the photographs are used to define volumes?

Section 4.3: a map is probably needed to illustrate this paragraph.

P11: lines 15-19: papers form Hungr can be cited.

Section 5.1: why to not present IDF to characterize rainfall and debris flow initiation or some other information about rainfall.

P 12: lines 13-16: what do you mean?

Table 2 caption: remind TLC meaning.

P14 lines 5-8: it is inconsistent.

P14 line 14: because of the boulders?

P14 lines 15-24: this is well known, you can find in NHESS paper about that (for instance Loye et al.)

Figure 8b: are sure that the integrals of the histograms have identical surfaces? If not why explain!

P16: instead of using deformation, it is better to use change of the bed topography or something similar. . . P16 line 17: how do you know that it is partially saturated?

Figure 10 and page 21 line 20: clarify the meaning of length ratio

P 20 line 13-15: I do not understand

P21 lines 14-19: what does mean exactly the percentage: gradient or proportion of something?

Page 22 line 5-8: fine sediment were not discussed before, why?

P 22 line 10: Meunier instead of Meunie

---

## Author Comment (AC2) · 20 Apr 2017

**Reply for review comments**

We sincerely thank you for the efforts you have made to improve our submission to *Natural Hazards and Earth System Sciences*. We have responded to all review comments in the following paragraphs. The blue-highlighted sentences are the review comments; sentences in black represent our responses to these review comments.

This study is an analysis is an attempt to characterize better the source zones of debris-flow in relationship with the storage slope angle and precipitations. The data were acquired in a catchment, which includes a large landslide. The monitoring was performed using TLS, video cameras, rainfall gauges and pressure sensor. In addition, field works were performed. The paper tries link saturation of the sediment, the volume of the initiation zones, the type of flow and slope angles. In conclusion, the flow characteristics be explained by the interplay of rainfall patterns saturating or not large or small volume of sediment. The slope gradient can be linked with the above conditions by the Takashi formulas. It shows that the fully saturated debris-flows create low gradient profile with breaks, while the partially saturated ones create constant steeper gradient.

Thank you for summarizing our paper.

General comments

The paper presents interesting results, but sometimes it is rather difficult to follow. It was difficult for me to write a summary, maybe the findings are not enough underlined and strengthened. In my opinion the authors have to read there paper carefully again simultaneously with a colleague that is not involved in the paper in order to clarifies make the text easier to read.

We will asked our colleague, who is not involved in this paper, to read our paper. In addition, we will improve "abstract", "introduction", and "summary and conclusion" to emphasize findings in our paper.

The abstract is as well no very informative.

The other reviewer also requested to revise abstract. We think statements on the findings were ambiguous. Therefore, we will revise the abstract to emphasize findings in our study.

The figures about the site are often too small and too dark.

We will changed lightness and contrast of the figures. In addition, we will extend the figures. The revised figure is shown in pg. 3 in this reply.

Information about rainfall are lacking such as IDF or other information. In addition, the relationship of the debris flows with the landslide is not really described.

We will added a figure showing initiation condition of the debris flow in the Ohya landslide. Please see details below written as replies for specific comments (pg. 6-8 in this reply). Although the initiation condition of the debris flow is basic information of the debris flow study, the target of our study is not the initiation condition, but the interaction between flow characteristics and accumulation condition of sediment storage. Therefore, we may not deeply discuss the initiation condition in the text.

In my opinion if the author clarify the text and make more easy to read this will be valuable paper about debris-flow behaviours.

Thank you for your comments improving our paper.

Specific comments
P2 line 1: "debris" instead of "decries"
We would like to remove the sentence based on the comment by the other reviewer.

P2 line 4: "Hungr" instead of "Hunger"
We should revise the misspelling.

P4 line 14: this must be explained.
When we apply Eq. (5) to the debris flow, porosity $n$ can be expressed as $1 - C$ using solid fraction $C$. The $n$ in the moving sediment mass becomes larger than the $n$ of storage when sediment particles start to disperse by collision with other particles (Hungr, 2005; Takahashi, 2014). We will add these explanations to the sentence. In addition, we would like to explain how we applied Eq. (5) to the debris flow by presenting the following equation, which explains relationship between the solid concentration in the steady-state flow (called equilibrium concentration) and the slope gradient is obtained (Takahashi, 1991; Egashira et al., 2001, Takahashi, 2014).

$$C = \frac{\gamma_w \tan\alpha}{(\gamma_s - \gamma_w)(\tan\phi - \tan\alpha)},$$

P4 lines 18-19: more explanations about dispersion
We would like to change the expressing for the better understanding by readers as follows:
"By substituting the porosity of the storage into $n$, Eq. (5) expresses the slope gradient needed for the entrainment of a fully saturated sediment mass, of which solid fraction is same as that of storage (hereafter referred to as $\alpha_2$).

Fig 1 caption: remind the letters meanings.

We will add explanation of the variables in the figure as follows:

"The $h$, $h_w$, and $z_1$ in the figure indicate heights at surface of the sediment mass, water table, and bottom of the sediment mass, respectively."

P6 line 10-14: to introduce this subject reference to Theule paper in NHESS can be introduced in the introductive section.

We think citation of Theule et al., 2012 is appropriate in "Introduction", rather than "study site". Thus, we would like to add the citation in the "Introduction".

Figure 2: limit of the landslide are missing or unclear.

We will changed lightness of the Figure 2 (please see below). We think boundary of the landslide is clear now. Lower end of the landslide is not obvious because more than three hundred years has been passed since initial failure of the landslide.

[Figure]

Figure 2

Page 7 line 13- p8: line 4: more explanations are needed to explain how these parameters are evaluated.

We will add detailed explanation on the analysis methods as follows.

"We identified timing of such debris flows based on changes in the topography observed by periodical photography with an interval of appropriately one week. Surface velocity of debris flows at 1 second intervals were obtained from time required for boulders on the flow surface to pass through fixed channel sections (2.0–5.0 m length) on the video images. The surface velocity provided by the video image analysis does not represent the flow depth averaged velocity. The mean velocity was estimated from surface velocity multiplied by 0.6, based on the velocity profile throughout the flows on movable beds obtained from a physically based model by Takahashi (1977, 2014). Flow depth of debris flows at 1 second intervals were also obtained from the video image analysis by reading level of the flow surface. Analysis points of the flow depth were set within the channel sections where changes in the channel bed topography attributed to occurrence of debris flows were minimum."

P8: this page is not well structured difficult to follow. For instance the ultrasonic sensor are used to measure the surface height are explained at the end but already introduced at the beginning of the page.

Flow depth obtained from video image analyses was used to calculate flow discharge. The flow depth obtained by the ultrasonic sensor was used to identify occurrence of debris flow as the backup of video cameras. We would like to added explanations to clarify aim of each observation method.

P8: line 23: where are installed the pressure sensor.

The water pressure sensor were installed at P3. We will clarify the location of installation site.

P9 lines 9-10: this accuracy was checked or it is it the manufacturer data?

This is data in the specifications. We will clarified source of the accuracy. The overall uncertainties of the point clouds, including scanning, registration, and georeferencing by GNSS, were considered in the order of centimeters to a decimeter. As noted at the end of this section, the accuracy of the measurement is explained in Hayakawa et al. (2016).

P10 line 4: only two target were used ???? not 3 minimum?

Unknowns needed for definition of a coordinate system to point clouds are x, y, z coordinates of the sensor, direction of x and y axes for rectangular coordinate system (two variables), depression angle of z axis (total six unknowns). Because the laser scanner was correctly leveled with its internal tilt sensor (giving an angle accuracy of 6''), the z axis is defined. Therefore, number pf unknowns in our study is four (coordinates of the sensor (x, y, z) and yaw angle (horizontal

direction)). The z coordinate of the sensor is readily obtained from that of one of the targets. The xy coordinates and yaw angle can be solved from the target xy coordinates by the 2-dimensional distance resection. We measured x, y, z coordinates at two targets (total six parameters). Therefore, number of parameters obtained by our measurement is sufficient to obtain values of unknowns.

In text, we will added an explanation that the scanner was correctly leveled with its internal tilt sensor.

We will revise the repeated sentence as follows:

"The two point clouds measured from different scan positions were registered using at least five reference targets placed between the two scan positions with accuracies of 0.5–6 mm."

Table 1: please add the point spacing of the cloud points.
We have added the point spacing in the Table 1 as follows. In addition, we will noted the point spacing in the text.

**Table 1: Date of TLS survey**

| Year | Date of survey | Number of debris flow after previous scanning | Date of last debris flow event[1] | Aveage point spacing of the cloud points (m) |
|------|---------------|---------------------------------------------|----------------------------------|--------------------------------------------|
| 2011 | November 11 | - | October 14 | 0.075 |
| 2012 | May 14 | 0 | - | 0.077 |
|      | August 23 | 2 | June 22 | 0.056 |
|      | November 21 | 3 | September 30 | 0.038 |
| 2013 | May 10 | 0 | - | 0.078 |
|      | August 16 | 0 | - | 0.040 |
|      | November 19 | 2 | September 15 | 0.057 |
| 2014 | May 16 | 0 | - | 0.094 |
|      | August 17 | 1 | August 10 | 0.066 |
|      | November 28 | 2 | October 5 | 0.085 |
| 2015 | May 15 | 0 | - | 0.094 |
|      | August 23 | 4 | August 17 | 0.083 |
|      | December 4 | 1 | September 9 | 0.130 |

[1] Date of the last debris flow event are not listed when debris flow had not occurred in the year , because the topography was possibly affected by the winter sediment supply rather than the last debris flow in the previous year.

As requested by the reviewer, we will noted the point density in the text. The point density ranged from 59.5 to 689.9 pts $m^{-2}$ (average 249.6 pts $m^{-2}$).

To clarify meaning of the sentence, we would like to revise the explanation on the mapping of

geomorphic units as follows:

"The extent of typical geomorphic units in the TLS survey area, including three rock slopes, three talus slopes, and a channel around the monitoring plot P1, was mapped by field surveys (Fig. 4). Distribution of the slope gradient with in these geomorphic units were calculated from TLS DEMs with various grid sizes. The mapping was conducted at the same time as each TLS survey because the area changed over time due to the sediment supply from outcrops and transportation of sediment by debris flows."

P11 line 5: how the photographs are used to define volumes?

The other reviewer also requested to clarify the method. We would like to add detailed explanation on the calculation method as follows:

"Photographs from site P4 cover the entire study site, whereas those from P1 focus on channel deposits at the bottom of the incised main channel, which is shaded in photographs from P4. By comparing these photographs with catchment topography and ortho photographs, which are obtained by airborne laser scanning (ALS) in seven periods (2005, 2006, 2009, 2010, 2011, 2012, and 2013), the area covered by storage (i.e., channel deposits and talus slopes) in each photograph periods was mapped on GIS. The bedrock topography in the upper Ichinosawa catchment was estimated from terrains by ALS in the periods when sediment storage was almost absent (i.e., in 2011 and 2012). Thirty-three cross-sectional areas of the storage with spacing of 25 m along the channels (24 cross-sections along the main channel and 9 cross-sections along a tributary) were calculated from the bedrock topography estimated from terrains by ALS and the location of both ends of the storage along the cross-sections on the GIS storage map (Fig. 2a) under the assumption that surface topography of the storage was an inclined line connecting both ends of the storage. Total volume of the storage was calculated by sum of the cross-sectional area multiplied by the spacing of cross-sections (25 m)."

Section 4.3: a map is probably needed to illustrate this paragraph.

In Figure 2, we will added cross-sectional lines used for estimation of the volume of storage. Please see revised Figure 2 shown above (pg. 3 in this reply). Cross-sectional lines are indicated as red lines.

P11: lines 15-19: papers form Hungr can be cited.

As suggested by the reviewer, we would like to cite Hungr et al. (2009) in the explanation of the errors in our analysis.

Section 5.1: why to not present IDF to characterize rainfall and debris flow initiation or some other information about rainfall.

As the reviewer pointed out, analysis on the rainfall threshold for occurrence of debris flow is essential for the debris flow studies. Thus, we would like to add a new figure showing rainfall

threshold for initiation of debris flow (please see following figure).

[Figure]

Figure: Comparison between rainfall duration and maximum 10-min rainfall intensity of rainfall events with total rainfall depth >2.0 mm and rainfall duration >10 min in the period from April 1998 to September 2015. Rainfall events with and without debris flows were potted using different markers. Size of the plot for debris flow events expresses proportional duration of partly saturated flow in overall debris flow surges. The rainfall intensity was calculated from rainfall data with a logging interval of 1 min. Rainfall events without 1-min interval data were not plotted. Dashed line, which can be expressed as Intensity = 8.86 (Duration)$^{-0.21}$, indicates lower limit of the rainfall condition needed for occurrence of debris flows when rainfall duration < 10 hr. Dash-dot line indicates the maximum rainfall intensity equals to 5 mm per 10-min, which is the rainfall threshold for occurrence of debris flow when the rainfall duration ≥10 hr.

We have also compared rainfall duration and average rainfall intensity during rainfall event (the total rainfall depth the divided by rainfall duration, please see below). Because initiation condition of debris flow in the Ohya landslide is highly affected by the short time rainfall intensity (10-min intensity), difference in the distribution of debris flow plots and non-debris flow plots were not clear in the IDF using average rainfall intensity (see the area surrounded by the red circle in the figure).

[Figure]

Based on suggestion by the other reviewer, we have revised the sentence as follows:

"The duration of each flow phase varied between the events. For example in the event of 5 August 2008, 88% of debris flow surges (percentage respect to the total event duration) was composed of partly saturated flow, while in the event of 30 August 2004 (Fig. 6), 90% in time of the phenomenon was composed of fully saturated flow."

We will explanation of TLC (time-lapse camera) in the Table 2 caption. We will also added meaning of TLC in Figure 3 caption.

We think the first half of the sentence in the previous version of the manuscript (location of the geomorphic units) was not needed. Therefore, we would like to remove the sentence as

follows:

"Mapping of the three geomorphic units (rock slopes, talus slopes, and the channel) in the TLS survey area showed that the size of each unit changed with time due to sediment supply and transport processes."

As the reviewer comments, the slope gradient decreases with increasing in the grid size because of the boulders. We will added statements relevant to the boulders as follows. "As reported by previous studies (Loye et al., 2009), the average slope gradient of each geomorphic unit becomes gentler with an increase in the grid size, because small scale asperities of terrains (e.g., boulders) are smoothed when the grid size is larger (Fig. 9a)."

P14 lines 15-24: this is well known, you can find in NHESS paper about that (for instance Loye et al.)

We would like to added citation of Loye et al. (2009) in the section. As the reviewer points out, the relationship between the slope gradient and the grid size is widely known, and is not novel. However, this part is basis of the discussion about the interaction between channel topography and debris flow type. Therefore, we would like to leave the section.

Figure 8b: are sure that the integrals of the histograms have identical surfaces? If not why explain!

Integrals of the histograms for each geomorphic unit are identical (=1). Therefore, histograms with different grid size (and different geomorphic units) are comparable. Because step of slope-gradient categories was 2 degree, integral of some histograms looks smaller than others. Thus, in the figure caption, we will explain that integrals of the frequency for each geomorphic unit in Fig. 8b, which are categorized with 2 degree step, are all 1.

P16: instead of using deformation, it is better to use change of the bed topography or something similar. . .

We agree that "changes of the channel bed topography" is easier for readers to image. Therefore, we will replace "channel bed deformation" to "changes in the channel bed topography" or other similar words. In some sentences, "changes in the channel bed topography" makes the sentence too long. Since "channel (or river) bed deformation" is also widely used in scientific papers, we would like to leave "channel bed deformation" in some sentences.

P16 line 17: how do you know that it is partially saturated?

We visually identified temporal changes in the flow type during debris flow events from

video images based on the existence of interstitial water on the flow surface. Video images just provide information at the flow surface. Therefore, we do not know thickness of the partly saturated layer. Nevertheless, based on the video image analysis, we can tell if the flow surface is saturated or unsaturated. We think the classification of the flow types was not easy for readers to image. Therefore, we will add video images of the two flow types as a new figure.

[Figure]

Figure: Images of fully and partly saturated debris flows captured by a time-lapse camera (TLC) at plot P2 in Fig. 2. (a) Fully saturated debris flow captured 9 September 2015, 8:41 (LT). (b) Partly saturated debris flow captured at 9 September 2015, 7:43 (LT). Cobbles and boulders covers flow surface of the partly saturated debris flow (Imaizumi et al., 2016b).

Figure 10 and page 21 line 20: clarify the meaning of length ratio

We will improve title of y-axis and caption of Figure 10. We also improved explanation on the length ratio at pg. 21, line 20 as follows:

"… the sum of the length ratio of the channel sections in the theoretical range of fully and partly saturated debris flows (between α1 and α2 ) in the active channel bed deformation zones was almost 1.0."

We would like to improve the sentence as: "our observations in the debris flow initiation zone of the upper Ichinosawa showed that overall debris flow surges were sometimes mainly composed of partly saturated flow".

P21 lines 14-19: what does mean exactly the percentage: gradient or proportion of something?

We will add a statement that the percentage is based on the slope gradient histogram.

Page 22 line 5-8: fine sediment were not discussed before, why?

We would like to delete the sentence based on the comment by the other reviewer. We think changes in the grain size distribution of channel deposits would be one factor affecting temporal changes in the standard deviation of the channel gradient, especially in the analyses with small grid size. However, we do not have quantitative data explaining temporal changes in the grain size distribution. In addition, as pointed out by the other reviewer, changes in the grain size of the channel deposits attributed to selective transport by the debris flow need deep discussion because that is not commonly recognized by the researchers. We think such additional discussion obscure target of this paper.

We will revise the citation.

---

## Author Response (AR1)

**Reply for review comments**

**Reviewer 1 (Prof. Carlo Gregoretti)**

We sincerely thank you for the efforts you have made to improve our submission to *Natural Hazards and Earth System Sciences*. We have responded to all review comments and have made appropriate modifications to our manuscript related to these comments as detailed in the following paragraphs. The blue-highlighted sentences are the review comments; sentences in black represent our responses to these review comments.

The authors present a study that link the debris flow types to the morphological characteristics of the initiation zone and the entrainment and deposition processes to the debris flow types. The writer identified the following deficiencies:

1) The text, in some parts, is not fluid but hard to read. It looks like an ensemble of pieces more and less linked each other.

We have checked entire manuscript again. In addition, we have asked our colleague, who is not involved in this paper, to read our paper. In order to improve flow of our paper, we have revised following points:

1. In the last paragraph of the introduction, we have described what we did in our study for better understanding of structure of the manuscript by readers (pg. 3, lines 6-12).
2. We have made a substantial revision in section 6.2, because the scope of the discussion was not clear in the previous version of the manuscript.
3. At the begging of some sections, we have added sentences explaining relationship with other part of the manuscript (pg. 3, lines 14-15; pg. 24, lines 16-20). We also improved topic sentences (pg. 16, lines 9-10; pg. 18, lines 9-10; pg. 23, lines 11-14).
4. We also revised "abstract" and "summary and conclusion", which are not informative in the previous version of the manuscript.
5. In the previous version of the manuscript, there was no statement on the grid size of GIS analysis, which is the main topic of chapter 5.2. Therefore, we have added statements on the grid size (pg. 2, lines 30-34).
6. We revised sections, which are not linked each other (e.g., explanation about the static model from pg. 4 line 8 to pg. 5 line 20).

2) Authors through a static equilibrium-based relationship (equation 5), obtain angles corresponding to different debris flow types or the sediment concentration of debris flow corresponding to bed slope angles, if the other quantities are known. If used for dynamic computation (i.e. the sediment volume concentration of the flowing material) this equation is misleading. In the case of motion, the equation (5) is a bit different (Lanzoni et al., 2017) and is obtained through the ratio between the basal bed shear stress and the basal normal stress. Moreover in the case of flowing material, the angle $\phi$ is not the static friction angle but the quasi-static of dynamic friction angle (Lanzoni et al., 2017). Therefore, also the sentence "Thus, not only…..Eq(5)", is uncorrected. Therefore, I suggest the authors to write that an equation of the same structure of eq. (5) can be obtained through the ratio between the basal bed shear stress and the basal normal stress that have a similar structure to the ration between shear stress and strength (Lanzoni et al., 2017).

Some governing equations for the debris flow in previous studies explained that only static frictional force (coulomb force), which can be obtained by the equations including volume concentration of the flowing material as a parameter, acts on the surface of the erosible bed (e.g., Egashira et al., 2001; Takahashi, 2014). However, as the reviewer pointed out, other studies consider flow dynamics differently. In any of these cases, we think the ratio between the shear stress and the normal stress at river bed is an important factor controlling flow characteristics. We have added the sentence suggested by the reviewer (pg. 4, lines 30-31). The sentence in p. 4 line 22-23 in previous version of the manuscript intended to mention that other types of sediment transport processes also satisfy the Eq. (5). We have revised the sentence to clarify what we mean (pg. 5, lines 2-3).

3) About triggering of debris flow, both the experiments of Gregoretti (2000) and the theoretical computations of Gregoretti (2008) clearly show that the entrainment of debris material into a water stream is provided by the surficial erosion of the debris layer rather than by the slide of the debris layer. Moreover, field works of Berti and Simoni (2005) , Cannon et al. (2008), Coe et al. (2008), Gregoretti and Dalla Fontana (2008), Theule et al. (2012), Hurliman et al. (2014), Degetto et al. (2015), Hu et al. (2016) point to a triggering mechanism where runoff erode the sediments and spread them along flow depth rather than the slide of a saturated mass. Recent works of Gregoretti et al. (2016) and Rengers et al. (2016) show that runoff descending from cliffs is an impulsive phenomenon characterized by a peaked hydrograph and that the impact of peak against debris deposits entrain enough material to have a solid-liquid current. Finally, authors cited the work of Kean et al. (2013) and Navratil et al. (2013). In the first work, it is expressly written that debris flow initiation by surface runoff is different from the debris flow initiation by shallow landslide (the title of the work deals with runoff-generated debris flow), while in the second it is stated that debris flow is initiated by surface runoff rather than landslide. Therefore, if the authors intend to use their slide model, they should state that channelized debris flow initiate by runoff as a surficial erosion (cite the references above) and that they approximate it by a "slide" model using the calculation that Prancevic et al. (2014) show.

Many debris flows in the Ohya landslide were also initiated by the erosion of the debris material by the surface runoff (Imaizumi et al., 2016). As the reviewer points out, erosion by the surface runoff is the predominant initiation mechanism of the debris flow. In the Ohya landslide, sliding of the debris material was also monitored. Our monitoring site is possibly in the traditional slope gradient range from fluvial processes to the failure of the debris deposits when we consider initiation mechanism of the debris flow like Prancevic et al. (2014). As the review suggested, we have added statements that many debris flows in the Ohya landslide and other debris flow torrents were triggered by the surface erosion. In addition, we noted that relationships between topography and type of sediment transport were discussed by approximation of the sediment transport type by the simple static models (pg. 5, lines 20-28).

Imaizumi, F., Tsuchiya, S. & Ohsaka, O. (2016) Field observations of debris-flow initiation processes on sediment deposits in a previous deep-seated landslide site, Journal of Mountain science, 13: 213. doi:10.1007/s11629-015-3345-9

About 6.1, the writer agrees with authors that sediment availability determines the type of debris flows but has some concern about partially saturated debris flow even if stated by other authors (e.g. Iverson and Vallance, 2001). They could be very dense debris flows, where fluid phase is just under the surficial sediments. Measurements carried out at Illgraben (Mc Ardell et al., 2007) show a flow density of the front that approximates that of a saturated terrain. An alternative way is that of very dense debris flow rich of debris material and more fluid debris flow. About the outcome of the correspondence between short-lasting rainfall and partially saturated debris flows and that between long-lasting rainfalls and saturated debris flows, the field experiences in the initiation area Cancia debris flow given by Bernard et al. (2016) and Gregoretti et al. (2016) seem to contradict it. At Rovina di Cancia (Northeastern Italian Alps) an hyperconcentrated flow occurred about 11 days later a partially saturated debris flows (according to the author definition; see the video at the following link https://www.youtube.com/watch?v=oKQSZVwOuRo). The rainfall depths were just a bit smaller in the second event even if the terrain was not dry as for the first event (Gregoretti et al., 2016). The main reason is that after the first event, channel did not recharge and the quantity of entrained sediments in the second event in the initiation reach was at least an order smaller than that entrained during the first event.

Thank you for telling us an interesting video in YouTube. As the reviewer comments, density of debris flow can be very high (close to that of saturated terrain) particularly at the head of surges even in the gentler channel (McArdell et al., 2007; McCoy et al., 2010; Okano et al., 2012). Unfortunately, we do not have data related to the internal structure of the debris flow (e.g., thickness of the unsaturated layer, depth profiles of solid concentration and flow velocity). Thus, we cannot deeply discuss dynamic mechanism of the flow. We have revised statements on the partly saturated debris flow in section 6.1 to clarify relationship between results in our study and previous studies (pg.23, lines 11-18). We have also discussed limitation in our analyses in the end of discussion (pg. 25, lines 23-29). As presented in Takahashi (2007) and Lanzoni et al. (2017), we think depth profile of the dynamic force affected by the slope gradient and the particle size relative to the flow depth should be considered to complete explanation of the dense debris flows. On the other hand, our simple static-force analyses and results of the field monitoring implied that unsaturated sediment transport frequently occur in the steep terrains. We think that point is worth to be emphasized in this paper.

We also agree with the reviewer's comment that recharge of the sediments into valley bottom affect volume of the sediment entrained by debris flows. Our monitoring results also implies that volume of debris flow material affects flow types (Fig. 7). In the end of the section 6.1, we have emphasized importance of the amount of the debris flow material and that of water supplied on the debris flow type (pg. 23, lines 10-14). Grain size, amount of debris flow material, and topography may affect difference in the flow characteristics and initiation mechanisms among debris flow torrents. In the end of the paper (in conclusion), we have added statements that we need to collect data in other debris flow torrents with different site conditions (e.g., grain size, slope gradient) for the further understanding of interactions between the debris flow characteristics and the topography (pg. 26, lines 22-26).

1. Page 1 – line 20: the sentence "The small-scale channel gradient…." is unclear.
We have revised the sentence. Statements on the channel gradient was added after the sentence (pg. 1, lines 23-25).

2. Page 1 - line 26: has been
We have replaced "have" to "has" (pg. 1, line 29).

3. Page 1 - line 28: the word "activities" after "monitoring" is missing.
As suggested by the reviewer, we have added "activities" (pg. 2, line 2).

4. Page 2 - line 1: some parts of the sentence "Understanding ………..system" are unclear.
We have revised the sentence. Now the sentence is "Thus, factors affecting debris-flow characteristics in the initiation zone (e.g., temporal changes in the solid fraction) is still unclear" (pg. 2, lines 4-5).

5. Page 2 – line 4: perhaps the reference Takahashi 2007 is better than Takahashi 1991.
We have replaced the reference "Takahashi, 1991" to "Takahashi, 2007" (pg. 2, line 7).

6. Page 2 - line 18: add the reference Gregoretti and Dalla Fontana (2008).
We have added the reference Gregoretti and Dalla Fontana (2008) (pg. 2, line 24).

7. Page 4 – line 5: what does it mean "is the exception ….. between α and ϕ?".
We have revised the explanation about pyroclastic flow. The relationship between α and ϕ expressed as Eq. (4) cannot be applicable to the pyroclastic flow (pg. 4, lines 12-18).

8. Page 4 – line 30: the sentence "The explanations…..2014)" is unclear as it regards the terrain formed by the sediment mass: what does it means? Moreover, the relationship given by Takahashi (2014) should be written.
The sentence explained relationship between solid fraction and channel gradient. In order to clarify the relationship between solid fraction and channel gradient, we have added Eq. (6). In addition, we replaced "terrain" to the "channel gradient" (pg. 5, lines 13-14).

9. Page 6 – lines 9-11: the sentence " Most of the channel…..low" is bad written and misleading.
This sentence explains the channel bed condition in the monitoring site. We have improved the sentence (pg. 7 line 2-4).

10. Page 7 – line 8: P2 is not in fig. 2a but in fig. 2b
We have removed "a" after "Fig. 2" (pg. 8, line 11).

11. Page 8 – lines 2 and 6: perhaps mean flow depth velocity or flow depth averaged velocity should be better than "mean velocity of all layers of the flow".
We have replaced "mean velocity of all layers of the flow" to "flow depth averaged velocity" (pg. 8, lines 19, 20).

12. Page 9 – line 14: the sentence "The location of …" is bad written.

We have revised the sentence. Now the sentence is "The location of this scan position varied with time because it was in the valley bottom where the topography changed due to erosion and deposition by debris flows."(pg. 11, lines 1-2).

We have added detailed explanation about the estimation method of the storage (pg. 12, lines 9-19).

We have replaced "section between sites" to "reach between the sites" (pg. 13, line 5).

We have replaced "high total rainfall" to "high value of total rainfall depth" (pg. 13, line 16).

We have replaced "of high intensity" to "characterized by high intensity" (pg. 13, line 17).

We have replaced "differed between events" to "varied between the events" (pg. 13, line 21).

We have rewritten the sentence as suggested by the reviewer (pg. 13, lines 21-23).

Originally we meant that difference in the channel gradient among camera locations potentially affects difference in the flow type. However, the channel gradient changes with time even at the same camera location. Therefore, now we think first half of the sentence was not needed (pg. 13, line 23).

We have rewritten the sentence. We think the sentence is easy to read now (pg.13, lines 23-26).

We think the term "roughness" was not appropriate. We have revised the sentence. We also explained how the slope gradient was calculated (pg. 16, line 11-13).

We have added explanations how to obtained DEMs from the TLS point clouds (pg. 11, line 21-23). We have also presented calculation method of the slope gradient (pg. 16, lines 11-12). We used the method proposed by Horn (1981).

We have added "the" in front of the "highest" (pg. 16, line 18).

We have rephrased the sentence. Now the phase is "a histogram of the slope gradient in talus slopes…" (pg. 16, line 22).

25. Page 15 – caption of Figure 7: calculated after using TLS data…..
We have revised the figure caption as suggested by the reviewer (pg. 17, new Fig. 8)

26. Page 16 – lines (1-8): the writer does not understand what the authors mean. In other words, what does it the meaning of the channel topography forming after debris flow occurrence?. It means that all the channel was flooded by debris flow or that the debris flow changed the bed.
We have added explanation about the aim of analyses on the longitudinal channel topography (pg. 18, lines 8-10). We tried to find out relationships between the flow type and the longitudinal profile of the channel (channel gradient) formed by the debris flow in the steep debris-flow initiation zone. Most part of the channel topography in the section between points A and E was changed by the debris flows on 30 September 2012 and 6 August 2015. In contrast, channel topography was not largely changed by the small debris flow on 17 August 2015. Therefore, we assumed that the channel profile on 23 August 2015 reflected characteristics of the debris flow on 6 August 2015, rather than that on 17 August 2015 (pg. 18, lines 16-17).

27. Page 16 – lines (9-14): even in these sentences the writer does not understand what the authors mean. Visual inspection of the two bed profiles of Figure 9a (are they from post-event surveys?) show that there is an high lowering of the bed profiles at the beginning and ending reaches. Authors should provide a much better description/comment of this figure.
We intended to explain similar characteristics as the reviewer comments. We have rewritten the paragraph (pg. 18, line 18-pg. 19, line 3). The topography was obtained by the post-event surveys (pg. 18, line 9).

28. Page 17 – caption of Figure 9: authors should specify that the profiles correspond to post-event surveys.
We have added timing of the debris flow events in the figure caption (pg. 20). Thus, now it is clear that the profile was measured after the debris flow events.

29. Pages 19-20. All the comments about α1, α2 the storage volume and slope gradient should consider that the middle part of the channel was not interested by erosion phenomena. Therefore, the authors should explain a reason and then exclude it from the further comments. Moreover, the sentence between pages 19 and 20 is not clear.
We have discussed the reason why the channel bed deformation was not clear in the middle part of the profile (pg. 19, lines 17-20. Channel was narrowed because of the massive rock cliffs at the left bank. Therefore, flow depth in this section may exceeds that in the other sections. High stream power attributed to the high flow depth may restricted deposition of the sediments, possibly resulting in the small channel bed deformation. We also excluded a sentence about inactive section. We revised the sentence which was between pages 19 and 20 (pg. 22, line 15-pg. 23, line 4).

30. Page 21 - lines 14-15. The explanation on the effect of grid size on hystogram shape should be coupled with some estimation of roughness (i.e. median grain size or something else) . Moreover, the authors should initiate the subsection explaining the reasons they produced the histograms.
We have added discussion between the shape of the histograms and grain size (pg. 24, lines 20-26). We think that the influence of the grain size on the slope gradient should be eliminated when we discuss relationship between the slope gradient and the sediment transport type (pg, 24, lines 26-28). Analyses of the histogram provides us idea on the influence of roughness attributed to the grain size on the slope gradient. We have added explanation about importance such analyses in the top of the section (pg. 24, lines 16-20).

31. Pages 21-22: the writer does not understand well the scope of the written sentences: authors should clearly rewrite them. For istance, fully saturated debris flow removed only fine sediments while partially saturated debris flows washed out entire bottom reaches. The writer thinks that the amount of entrained material depend also on the main characteristics of the of the solid-liquid current: flow depth, sediment concentration and velocity that in the case of coarse grained debris flow depends also on the runoff discharge (Lanzoni et al., 2017).
We agree with the reviewer's comment that the amount of entrained material depends on the flow depth, sediment concentration and velocity (this can be expressed as a function of flow depth and sediment

concentration). In this section, we did not discuss amount of entrained material, but we discussed relationship between the (channel) topography and the type of the sediment transport. In order to make the topic clear, we have changed title of the section to "Influence of the sediment transport type on the slope gradient of terrains". The topic about the selective transport of the channel deposits would be not necessary in this paper. In addition, that topic may have obscured scope of the discussion. Therefore, we have removed the statements on the selective transport. We also revised the end of the section to emphasize importance of the relationships between the channel topography and type of sediment transport (pg. 25, lines 28-29).

**Reviewer 2 (Prof. Michel Jaboyedoff)**

We sincerely thank you for the efforts you have made to improve our submission to *Natural Hazards and Earth System Sciences*. We have responded to all review comments in the following paragraphs. The blue-highlighted sentences are the review comments; sentences in black represent our responses to these review comments.

This study is an analysis is an attempt to characterize better the source zones of debris-flow in relationship with the storage slope angle and precipitations. The data were acquired in a catchment, which includes a large landslide. The monitoring was performed using TLS, video cameras, rainfall gauges and pressure sensor. In addition, field works were performed. The paper tries link saturation of the sediment, the volume of the initiation zones, the type of flow and slope angles. In conclusion, the flow characteristics be explained by the interplay of rainfall patterns saturating or not large or small volume of sediment. The slope gradient can be linked with the above conditions by the Takashi formulas. It shows that the fully saturated debris-flows create low gradient profile with breaks, while the partially saturated ones create constant steeper gradient.

Thank you for summarizing our paper.

General comments

The paper presents interesting results, but sometimes it is rather difficult to follow. It was difficult for me to write a summary, maybe the findings are not enough underlined and strengthened. In my opinion the authors have to read there paper carefully again simultaneously with a colleague that is not involved in the paper in order to clarifies make the text easier to read.

We have read paper carefully again. In addition, we have asked our colleague, who is not involved in this paper, to read our paper. We have revised our manuscript to emphasize our findings.

1. In the last paragraph of the introduction, we have emphasized what we did in our study for better understanding of structure of the manuscript by readers (pg. 3, lines 6-12).
2. We have made a substantial revision in section 6.2, because the scope of the discussion was not clear in the previous version of the manuscript.
3. We have improved topic sentences (pg. 16, lines 9-10; pg. 18, lines 9-10; pg. 23, lines 11-14).
4. In order to emphasize our findings, we also revised "abstract" and "summary and conclusion".
5. We have deleted discussions that are not closely related to the target of our study.

The abstract is as well no very informative.

The other reviewer also requested to revise abstract. We think statements on the findings were ambiguous. Therefore, we have revised the abstract to emphasize findings in our study (pg. 1, lines 18-25).

The figures about the site are often too small and too dark.

We have changed lightness and contrast of the figures (Figs. 2, 7). In addition, we have extended the figures.

Information about rainfall are lacking such as IDF or other information. In addition, the relationship of the debris flows with the landslide is not really described.

We have newly added a figure showing initiation condition of the debris flow in the Ohya landslide (Figure 5). Please see details below written as replies for specific comments. Although the initiation condition of the debris flow is basic information of the debris flow study, the target of our study is not the initiation condition, but the interaction between flow characteristics and accumulation condition of sediment storage. Therefore, we did not deeply discuss the initiation condition.

In my opinion if the author clarify the text and make more easy to read this will be valuable paper about debris-flow behaviours.

Thank you for your comments improving our paper.

Specific comments

P2 line 1: "debris" instead of "decries"

This sentence has been removed based on the comment by the other reviewer.

P2 line 4: "Hungr" instead of "Hunger"
We have revised the misspelling (pg. 2, line 7).

P4 line 14: this must be explained.
We have added explanation to the sentence. In addition, we have explained how we applied Eq. (5) to the debris flow by presenting Eq. (6) (pg. 4, lines 23-31).

P4 lines 18-19: more explanations about dispersion
We have changed the expression for the better understanding by readers. The expression is now "of which solid fraction is same as that of storage" (pg. 5, lines 4-5).

Fig 1 caption: remind the letters meanings.
We have added explanation of the variables in the figure caption.

P6 line 10-14: to introduce this subject reference to Theule paper in NHESS can be    introduced in the introductive section.
We think citation of Theule et al., 2012 is appropriate in "Introduction", rather than "study site". Thus, we have added its citations in the "Introduction" (pg. 2, lines 25, 27).

Figure 2: limit of the landslide are missing or unclear.
We have changed lightness of the Figure 2. Therefore, we think boundary of the landslide is clear now. Lower end of the landslide is not obvious because more than three hundred years has been passed since initial failure of the landslide (Figure 2).

Page 7 line 13- p8: line 4: more explanations are needed to explain how these parameters are evaluated.
We have added detailed explanation on the analysis methods (pg. 8, lines 17-24).

P8: this page is not well structured difficult to follow. For instance the ultrasonic sensor are used to measure the surface height are explained at the end but already introduced at the beginning of the page.
Flow depth obtained from video image analyses was used to calculate flow discharge. The flow depth obtained by the ultrasonic sensor was used to identify occurrence of debris flow as the backup of video cameras. We have added explanations to clarify aim of each observation method (pg. 8, lines 21-16; pg. 9, lines 9-11).

P8: line 23: where are installed the pressure sensor.
The water pressure sensor were installed at P3. We have clarified the location of installation site (pg. 9, line 6).

P9 lines 9-10: this accuracy was checked or it is it the manufacturer data?
This is the data in the specifications. We have clarified source of the accuracy (pg. 10, lines 8-9). The overall uncertainties of the point clouds, including scanning, registration, and georeferencing by GNSS, were considered in the order of centimeters to a decimeter (pg. 11, lines 9-10). As noted at the end of this section, the accuracy of the measurement is explained in Hayakawa et al. (2016).

P10 line 4: only two target were used ???? not 3 minimum?
Unknowns needed for definition of a coordinate system to point clouds are x, y, z coordinates of the sensor, direction of x and y axes for rectangular coordinate system (two variables), depression angle of z axis (total six unknowns). Because the laser scanner was correctly leveled with its internal tilt sensor (giving an angle accuracy of 6''), the z axis can be defined. Therefore, number of unknowns in our study is four (coordinates of the sensor (x, y, z) and yaw angle (horizontal direction)). The z coordinate of the sensor is readily obtained from that of one of the targets. The xy coordinates and yaw angle can be solved from the target xy coordinates

by the 2-dimensional distance resection. We measured x, y, z coordinates at two targets (total six parameters). Therefore, number of parameters obtained by our measurement is sufficient to obtain values of unknowns. We have added an explanation that the scanner was correctly leveled with its internal tilt sensor (pg. 10, lines 11-12).

P10 first lines are repeated.
We have revised the repeated sentences (pg. 11, lines 2-4).

Table 1: please add the point spacing of the cloud points.
We have added the point spacing in the Table 1. We also noted the point spacing in the text (pg. 11, lines 4-5).

P10 line 10: original density of points is needed here
As requested by the reviewer, we have noted the point density in the text (pg. 11, lines 4-5).

P10 lines 21-23: unclear please clarify.
We have revised the explanation on the mapping of geomorphic units. We think the phases in the revised manuscript are clearer than those in previous version of the manuscript (pg. 11, line 21-pg. 12, line 2).

P11 line 5: how the photographs are used to define volumes?
We have added detailed explanation on the calculation method of the debris flow volume (pg. 12, lines 9-19).

Section 4.3: a map is probably needed to illustrate this paragraph.
In Figure 2, we have added cross-sectional lines used for estimation of the volume of storage.

P11: lines 15-19: papers form Hungr can be cited.
As suggested by the reviewer, we have cited Hungr et al. (2009) in the explanation of the errors in our analysis (pg. 13, lines 1-2).

Section 5.1: why to not present IDF to characterize rainfall and debris flow initiation or some other information about rainfall.
As the reviewer pointed out, analysis on the rainfall threshold for occurrence of debris flow is essential for the debris flow studies. Thus, we have newly added a figure showing rainfall threshold for initiation of debris flow (Figure 5). However, this paper already includes many topics, including discussion on the static force, evaluation of the analysis methods, relationship between window size and analyzed topography, debris flow characteristics, and temporal changes in the topography in the debris flow torrent. Because this paper focuses on interaction between characteristics of debris flows and accumulation condition of the storage, we did not deeply discuss initiation condition of the debris flow. Instead, we presented relationship between rainfall pattern on IDF and the flow type (Figure 5).
We have also compared rainfall duration and average rainfall intensity during rainfall event (the total rainfall depth the divided by rainfall duration, see below). Because initiation condition of debris flow in the Ohya landslide is highly affected by the short time rainfall intensity (10-min intensity), difference in the distribution of debris flow plots and non-debris flow plots were not clear in the IDF using average rainfall intensity (see

the area surrounded by the red circle in the figure).

[Figure]

P 12: lines 13-16: what do you mean?
Based on suggestion by the other reviewer, we have revised the sentence. Now the sentence is "The duration of each flow phase varied between the events. For example in the event of 5 August 2008, 88% of debris flow surges (percentage respect to the total event duration) was composed of partly saturated flow, while in the event of 30 August 2004 (Fig. 6), 90% in time of the phenomenon was composed of fully saturated flow." (pg. 13, lines 20-23).

Table 2 caption: remind TLC meaning.
We have added explanation of TLC (time-lapse camera) in the Table 2 caption. We have also added meaning of TLC in Figure 3 caption.

P14 lines 5-8: it is inconsistent.
We think the first half of the sentence in the previous version of the manuscript (location of the geomorphic units) was not needed. Therefore, we have removed that part (pg. 16, line 9).

P14 line 14: because of the boulders?
As the reviewer comments, the slope gradient decreases with increasing in the grid size because of the boulders. We have added statements relevant to the boulders (pg. 16, lines 20-21).

P14 lines 15-24: this is well known, you can find in NHESS paper about that (for instance Loye et al.)
We have newly added citation of the Loye et al. (2009) in the section (pg. 16, line 19). As the reviewer

points out, the relationship between the slope gradient and the grid size is widely known, and is not novel. However, this part is basis of the discussion about the interaction between channel topography and debris flow type. Therefore, we left the section in the revised manuscript (pg. 16, line 19-pg. 17, line 3).

Figure 8b: are sure that the integrals of the histograms have identical surfaces? If not why explain!
Integrals of the histograms for each geomorphic unit are identical (=1). Therefore, histograms with different grid size (and different geomorphic units) are comparable. Because step of slope-gradient categories was 2 degree, integral of some histograms looks smaller than others. Thus, we have explained in the figure caption.

P16: instead of using deformation, it is better to use change of the bed topography or something similar. . .
We agree that "changes of the channel bed topography" is easier for readers to image. Therefore, we have replaced "channel bed deformation" to "changes in the channel bed topography" or other similar words. In some sentences, "changes in the channel bed topography" makes the sentence too long. Since "channel (or river) bed deformation" is also widely used in scientific papers, we left "channel bed deformation" in some sentences.

P16 line 17: how do you know that it is partially saturated?
We visually identified temporal changes in the flow type during debris flow events from video images based on the existence of interstitial water on the flow surface (pg. 8 lines 33-34). Video images just provide information at the flow surface. Therefore, we do not know thickness of the partly saturated layer. Nevertheless, based on the video image analysis, we can tell if the flow surface is saturated or unsaturated. We have improved the explanation (pg. 8, 30-31).

Figure 10 and page 21 line 20: clarify the meaning of length ratio
We have revised title of y-axis and caption of new Figure 11. We also improved explanation on the length ratio at pg. 25, lines 1-3.

P 20 line 13-15: I do not understand
We have improved the sentence. Now the sentence is "our observations in the debris flow initiation zone of the upper Ichinosawa showed that overall debris flow surges were sometimes mainly composed of partly saturated flow" (pg. 23, lines 12-14).

P21 lines 14-19: what does mean exactly the percentage: gradient or proportion of something?
We have clarified that the percentage is based on the slope gradient histogram (Fig. 9b) (pg.24, lines 29-34).

Page 22 line 5-8: fine sediment were not discussed before, why?
We have deleted the sentence based on the comment by the other reviewer. We think changes in the grain size distribution of channel deposits would be one factor affecting temporal changes in the standard deviation of the channel gradient, especially in the analyses with small grid size. However, we do not have quantitative data explaining temporal changes in the grain size distribution. In addition, as pointed out by the other reviewer, changes in the grain size of the channel deposits attributed to selective transport by the debris flow need deep discussion because that is not commonly recognized by the researchers. We think such additional discussion obscure target of this paper. Therefore, we have deleted the sentence.

P 22 line 10: Meunier instead of Meunie
We have revised the citation (pg. 25, line 23).

[revised manuscript text omitted]

---

## Referee Report (RR1)

**RE:** NHESS 2017 20        Imaizumi et al. Interactions between the accumulation of sediment storage and debris flow characteristics in a debris-flow initiation zone, Ohya landslide body, Japan

**Overview**

The authors improved their work but there are still some deficiencies:

1) The work, even if valuable, is yet not well explained. The paper, even if improved in some parts, is still hard to read. A clear or direct focus is missing and the parts seem untied. As a example, the analyze of the bed profile should be directly linked to the proof and determination of the angles $\alpha_i$. Moreover, the paper should be written in a more concise form.

2) The writer does not the share the authors approach on the limiting slope angles because it is uncorrected and misleading as it is shown.

   The triggering of debris flow along channel is mostly due to runoff. Unfortunately, as we do not know runoff, we cannot incorporate it in the scheme based on a static equilibrium-based relationship (equation 5). Therefore, the justifications of the authors appears physically uncorrected.

   Moreover, you can have debris flow with slope angle smaller than that given by equation (5) due to runoff. I suggest the authors to rewrite this part, justifying their approach with the impracticality of introducing the runoff contribute in the equilibrium relationship. The limiting inferior angles could be that given by flume laboratory experiments (15° according to Takahashi, 1978).

3) The second part of the text (from subsection 4.3), is again not fluid but very hard to read. Moreover, because in some parts it is too detailed, it is not readable.

4) In subsection 4.3 the writer does not understand the technique used for the topographical measurement? Photographs or ALS. This part should be rewritten in a clear form. Moreover, what is it the meaning of **"estimated from terrain by ALS"**?. Finally, I suggest the authors to initially and briefly summarize all the techniques they used and the scope.

5) In subsection 5.3 analyzing the bed profile after two events the authors observe that in the medium part it remains unchanged. They explain this by the absence of deposition. A consequence of this explanation is that in the upstream and downstream reaches there is deposition. This fact means that the considered reach is not the initiation area where

entrainment should occur rather deposition. In this case the scheme proposed by the authors at section 2 cannot be applied here.

The following are the detailed comments and specifications.

1. Page 1 – line 12: perhaps usually is better than often.
2. Page 1 - line 23: The sentence "The small-scale channel…." should be rewritten. At the beginning it should be specifiy that debris flows form some channel within the bed. If the writer has understanden what the authors mean.
3. Page 2 - line 22: the reference Gregoretti and Dalla Fontana, (2008) should be placed here and not at line 26.
4. Page 4 – line 15: terrain with a slope smaller than is better.
5. Page 4 - lines 15-19: this sentence could be neglected.
6. Page 5 – line 5: the reference Gregoretti, 2000 could be substituted by Gregoretti, 2008.

7. Page 7 – line 2: perhaps it is better "is" rather than "was";
8. Page 7 – line 2: place there is before a
9. Page 7 – line 3: place is after bedrock
10. Page 7 – line 7: decreased?
11. Page 8 – line 9: please mark the position of the monitoring station on the map.
12. Page 8 – line 11: then moved upstream to P2 is better.
13. Page 9 – line 13: is runoff increasing or decresing slowly?
14. Page 19 – lines 19-20: In this sentence it is stated that in the middle part the bed is unchanged because there is no deposition. This indirectly states that changes in the upper and lowest are only due to deposition. In other words in the upper and lower reaches there is only deposition and no entrainment. Is this correct? Authors should introduce this information at the beginning of the subsection.
15. Page 23 – lines 1-8: Confused sentences: the writer does not understand their meaning.
16. Page 23 – line 13: overall and sometimes cannot be in the same sentence that results confused.
17. Page 23 – lines 19-20. This sentence contradicts what written at line 25 of page 13.
18. Pag. 24 – lines 3-7. The writer agrees with the authors that the amount of storage influence the type of debris flow front (partially or fully saturated) but not the rainfall. In Rovina di

Cancia, in the summer two debris flows were triggered by two convective storms of nearly equal duration. The front of the first debris flow was partially saturated while the second was fully saturated (Gregoretti et al., 2016). This means that rainfall type in the determination of the front type could be secondary respect to the available storage volume.

Caption of Figure 1: please substitute water table with water level.

Gregoretti C., Degetto M., Bernard M., Crucil, G., Pimazzoni A., De Vido G., Berti M., Simoni   A. Lanzoni S. Runoff of small rocky headwater catchments: Field observations and hydrological modeling. *Water Resources Research*. 52(8) doi: 10.1002/2016WR018675

---

## Referee Report (RR2)

Dear Editor,

Please find here below my second review of the paper nhess-2017-20 R1:

Interactions between the accumulation of sediment storage and debris flow characteristics in a debris-flow initiation zone, Ohya landslide body, Japan

By

Fumitoshi Imaizumi, Yuichi S. Hayakawa, Norifumi Hotta, Haruka Tsunetaka, Okihiro Ohsaka, Satoshi Tsuchiya

The paper has been greatly improved, the authors made it easier to follow. Nevertheless, some points have to be clarified:

• As $\alpha$ 1 and 2 are described in section 2, it will be nice to remind there meaning later within text, or change their name like: $\alpha_d$ (for dry) and $\alpha_{fs}$ (for fully saturated), this will be easier for the reader to remember.

• In figure 2, I still do not understand where the Ohya landslide is? If I look at the figure landslide = catchment area, which is strange for me.

• P11 line 17: linear instead of liner?

• Legend figure 6: "hydrograph" appears only here in the caption could you make the link with the text (ultrasonic sensor?) and reminds where it was located P1 or p2?

• P18 section 5.3: In my opinion, you must clarify about the term "Bed profile", "longitudinal channel profile". To which type of profile do you refer always the longitudinal one?

• Table 3: the gradient are in degrees. If so please add the "°"

If the the authors solved these problems in my opinion the paper will be a nice contribution to the scientific debate on debris-flow initiation.

---

## Referee Report (RR3)

**RE:** NHESS 2017 20 Imaizumi et al. Interactions between the accumulation of sediment storage and debris flow characteristics in a debris-flow initiation zone, Ohya landslide body, Japan

**Overview**

The authors improved their work but there are some parts to be refined:

1) The subsection 4.3 is not understandable. The first sentence is not clear: what does it mean "We estimated the volume of storage from periodical photography and terrains …". For the following part, I suggest the authors to write that 36 cross section lines were considered along the reach and volume of storage was computed for each of the 35 areas between the cross-section line. Moreover, at the beginning of the second sentence did the authors confuse storage for channel?

2) About debris flow monitoring (5.1), the reviewer does not understand the different separation time of rainfall durations for the two thresholds, 10 h, when the rainfalls are classified in two groups based on rainfall duration smaller or larger than 5h. Moreover, legend of Figure 5 should be corrected: no debris flow instead of debirs flow. The unknown points of Figure 5 need some specification or explanation in the caption or in the text: do they correspond to fully saturated debris flows or both to partly and fully unsaturated debris flows? About the weak positive dependence of partly saturated debris flow on the storage volume, a reason could be the dependence from another factor: the precipitated depth before debris flow occurrence (see point 4).

3) About subsection 5.3, where are the points F and G on Figure 2?

4) About section 6.1, the sentence at the lines 16-19 of page 21 is not very clear: could you rewrite? Moreover, the writer agrees with the authors that the storage volume can influence the debris flow type. About the two debris flows occurred at Cancia in summer of year 2015 (Gregoretti et al., 2016), the former was partly saturated in the first minute while the latter was fully saturated. The reason is that the upstream storage volume in the second case was negligible because washed out by the first event occurred 12 days before. Authors, if willing, could insert this fact in this subsection. Considering Figure 7, another factor controlling the debris flow type could be the rainfall precipitated before the debris flow occurrence in the case of long duration rainfalls (typhoons). In this case the storage volume could be partly or entirely saturated by the rainfall previous debris flow and the degree of

saturation could influence the proportional amount of partly saturated debris flow. Authors could explore this possibility after analyzing the rainfall data.

5) In section 7, authors should add some specification to support the sentence "In addition, our study elucidated that the slope geomorphic units is the key factor in the estimation of predominant type of the sediment transport…"

The following are the detailed comments and specifications.

1. Page 5 – line 9: ratio instead of ration?
2. Page 22- line 27: what is it the length ratio of the channel sections? (see also Figure 11)
3. Page 23– line 20: please insert could before exist.
4. Page 24 - line 27: is instead of are after Paolo Tarolli.

Gregoretti C., Degetto M., Bernard M., Crucil, G., Pimazzoni A., De Vido G., Berti M., Simoni A. Lanzoni S. Runoff of small rocky headwater catchments: Field observations and hydrological modeling. *Water Resources Research*. 52(8) doi: 10.1002/2016WR018675

---

## Author Response (AR2)

**Reply for review comments**

Reviewer 1 (Prof. Carlo Gregoretti)

We sincerely thank you for the efforts you have made to improve our submission to *Natural Hazards and Earth System Sciences*. As replied later, we have removed many sections that were not very important in this paper. We think focus of the revised paper is clearer than that of previous version of the paper. We have responded to all review comments and have made appropriate modifications to our manuscript related to these comments as detailed in the following paragraphs. The blue-highlighted sentences are the review comments; sentences in black represent our responses to these review comments.

**Overview**
The authors improved their work but there are still some deficiencies:
1) The work, even if valuable, is yet not well explained. The paper, even if improved in some parts, is still hard to read. A clear or direct focus is missing and the parts seem untied. As a example, the analyze of the bed profile should be directly linked to the proof and determination of the angles $\alpha i$. Moreover, the paper should be written in a more concise form.

The linkage between theoretical channel gradients ($\alpha_1$ and $\alpha_2$) and real channel gradient was presented in the last part of the section 5.3. However, as pointed out in comment 15, statements on the comparison between theoretical channel gradients and real channel gradient was not clear. We have rewritten the section. Now statement is much clearer than the previous version of the manuscript (pg. 20, line 14 - pg. 21, line 9). In addition, as replied to comment 3), we have removed many sentences to focus our results and discussion on the temporal changes in the channel gradient.

2) The writer does not the share the authors approach on the limiting slope angles because it is uncorrected and misleading as it is shown. The triggering of debris flow along channel is mostly due to runoff. Unfortunately, as we do not know runoff, we cannot incorporate it in the scheme based on a static equilibrium-based relationship (equation 5). Therefore, the justifications of the authors appears physically uncorrected.
Moreover, you can have debris flow with slope angle smaller than that given by equation (5) due to runoff. I suggest the authors to rewrite this part, justifying their approach with the impracticality of introducing the runoff contribute in the equilibrium relationship. The limiting inferior angles could be that given by flume laboratory experiments (15° according to Takahashi, 1978).
We also think that many debris flows are triggered by surface runoff. As the reviewer points out, debris flows do not satisfy solid concentration expressed by equations based on static force, especially in their initial stage of the development. We agree that sediment transport type is not completely determined by the slope gradient. At the same time, even if our model is not strict and sometimes incorrect, we think it

is important to apply simple models to complex debris flow in the field when we figure out overall processes in the debris flow initiation zone. Before we start to discuss debris flow, we have explained the reason why we apply a simple static model (p.4, lines 16-28). We also added statements on the lower boundary of the channel gradient that the equation for equilibrium concentration of debris flow (Eq. (6)) is applicable (pg. 5, lines 9-13).

3) The second part of the text (from subsection 4.3), is again not fluid but very hard to read. Moreover, because in some parts it is too detailed, it is not readable.

We have removed sentences which were not closely related to focus of this paper. Location of removed sentences are highlight green in the manuscript (pg. 11, line 18; pg. 15, line 8; pg. 15, line 9; pg. 15, line 11; pg. 17, line 17; pg. 22, line 4, pg. 22, line 7; pg. 22, line 14; pg. 23, line 8; pg. 23, line 10). In addition, we reconstructed structure of the sections 4.3 and 5.3.

4) In subsection 4.3 the writer does not understand the technique used for the topographical measurement? Photographs or ALS. This part should be rewritten in a clear form. Moreover, what is it the meaning of **"estimated from terrain by ALS"**?. Finally, I suggest the authors to initially and briefly summarize all the techniques they used and the scope.

We have rewritten section 4.3. Explanation of the method has been summarized. "estimated from terrain by ALS" meant that bedrock topography under the storage was estimated from terrains obtained by ALS in 2011 and 2012 when sediment storage was almost absent (pg. 11, lines 13-18).

5) In subsection 5.3 analyzing the bed profile after two events the authors observe that in the medium part it remains unchanged. They explain this by the absence of deposition. A consequence of this explanation is that in the upstream and downstream reaches there is deposition. This fact means that the considered reach is not the initiation area where entrainment should occur rather deposition. In this case the scheme proposed by the authors at section 2 cannot be applied here.

The channel section, in which longitudinal profile of the channel were analyzed, is the initiation zone of debris flow, because initial movement of the sediment mass (debris flow) have been monitored by video cameras and TLCs in this section. Both erosion and deposition occurs in the lowermost and uppermost reaches by the occurrence of debris flows. Sediment supply from hillslopes also cause deposition in these sections. We have added these information at the beginning of the subsection (pg. 17, lines 10-13). We also improve the sentence pg. 17, line 29 to prevent misunderstanding by readers.

The following are the detailed comments and specifications.
1. Page 1 – line 12: perhaps usually is better than often.
As suggested by the reviewer, we have replaced often with usually (pg. 1, line 12).

2. Page 1 - line 23: The sentence "The small-scale channel…." should be rewritten. At the beinning it

should be specifiy that debris flows form some channel within the bed. If the writer hasg understanden what the authors mean.

We have revised the sentence. We hope the meaning of the sentence is clear now (pg.1, line 22).

3. Page 2 - line 22: the reference Gregoretti and Dalla Fontana, (2008) should be placed here and not at line 26.

We have moved the reference to the former sentence. (pg. 2 line 21)

4. Page 4 – line 15: terrain with a slope smaller than is better.

We have revised as suggested by the reviewer (pg. 4, line 15).

5. Page 4 - lines 15-19: this sentence could be neglected.

We have removed the sentence (pg. 4, line 15).

6. Page 5 – line 5: the reference Gregoretti, 2000 could be substituted by Gregoretti, 2008.

We think the reviewer intended to point out the reference in pg. 5, line 23 in the previous version of the manuscript. We have substituted the reference (pg 4, line 19).

7. Page 7 – line 2: perhaps it is better "is" rather than "was";

We have replaced "was" with "is" (pg. 7, lines 1)

8. Page 7 – line 2: place there is before a

We have added "there is" in the text (pg. 7, lines 1-2)

9. Page 7 – line 3: place is after bedrock

We have added "is" after bedrock (pg. 7, line 2)
.

10. Page 7 – line 7: decreased?

We also think "decreased" is better. We have revised (pg. 7, line 6).

11. Page 8 – line 9: please mark the position of the monitoring station on the map.

Sensors, cameras, and data loggers were not located at one specific site, but located one (or two) sites in the section between P1 to P3. We specified that in the first sentence of the section (pg. 8, lines 3-4). Location of each sensors are specified in the explanation of each sensor in section 4.1.

12. Page 8 – line 11: then moved upstream to P2 is better.

We have added "upstream" after "moved" (pg. 8, line 5).

13. Page 9 – line 13: is runoff increasing or decresing slowly?

Runoff (water height and water pressure) without debris flow increased and decreased slowly (over ten minutes to several hours), while that during debris flow events abruptly changes within 1 minute (Fig. 6). We have revised the sentence to specify such changes (pg. 9, line 5).

In upper and lower reaches, active erosion, deposition and entrainment of sediment occur. We have added such information in the first paragraph of the subsection (pg. 17, lines 10-13). We have replaced "deposition" with "accumulation of storage" to prevent misunderstanding by readers (pg. 17, line 29).

We have rewritten the sentences by simple words (pg. 20, line 14- pg. 21, line 9). We hope the sentences are clear now.

We have removed "overall" in front of "debris flow" (pg. 21, line 14).

We have changed the sentence pg. 23 line 19 in previous version of the manuscript to the same expression as pg. 13 line 25 (pg. 21, line 20).

The sentence in pg 24 line 3-5 in previous version of the manuscript is not our finding, but results reported in Okano et al. (2012). We have revised the sentence to make it clear (pg. 22, lines 4-6). Some studies have pointed out relationship between rainfall patterns and flow characteristics (Okano et al., 2012; Kean et al., 2013; Hürlimann et al., 2014). At the same time, as pointed out by the reviewer, different flow types appear even if the pattern of rainfall is similar. Therefore, we have revised the sentence to prevent misunderstandings (pg. 22, lines 4-7). In addition, we have removed sentence in pg. 24 lines 6-11 (in the previous version of the manuscript), which explained how rainfall pattern affects the debris flow type.

Caption of Figure 1: please substitute water table with water level.

We have replaced "table" with "level" as suggested by the reviewer (Fig. 1 caption).

We sincerely thank you for the efforts you have made to improve our submission to *Natural Hazards and Earth System Sciences*. We have responded to all review comments and have made appropriate modifications to our manuscript related to these comments as detailed in the following paragraphs. The blue-highlighted sentences are the review comments; sentences in black represent our responses to these review comments.

The paper has been greatly improved, the authors made it easier to follow. Nevertheless, some points have to be clarified:

• As α1 and α2 are described in section 2, it will be nice to remind there meaning later within text, or change their name like: αd (for dry) and αfs (for fully saturated), this will be easier for the reader to remember.

Thank you for your suggestion. $\alpha_2$ is the boundary between partly and fully saturated flows. Readers may misunderstand that the slope gradient relates to just one of the two flow type if we add suffix of a specific flow type. Thus, we have reminder of the meaning in the latter half of the paper (pg. 20, line 10, pg. 21, lines 2, 3, 16).

• In figure 2, I still do not understand where the Ohya landslide is? If I look at the figure landslide = catchment area, which is strange for me.

Ohya landslide is a huge landslide with a length and a width of 1.8 and 1 km, respectively. The landslide is surrounded by black solid line in Fig. 2a. We have noted that in figure caption.

In addition, we have stated that Ohya-landslide is currently composed of some sub-catchments (pg. 6, line 14-15).

• P11 line 17: linear instead of liner?

We have replaced "liner" with "linear" (pg. 10, line 22)

• Legend figure 6: "hydrograph" appears only here in the caption could you make the link with the text (ultrasonic sensor?) and reminds where it was located P1 or p2?

The hydrograph is obtained from video camera images. We have noted location of video cameras in the figure captions (Fig. 6). We have added statements on the hydrograph in the text, as suggested by the reviewer (pg. 12, lines 19-20).

• P18 section 5.3: In my opinion, you must clarify about the term "Bed profile", "longitudinal channel profile". To which type of profile do you refer always the longitudinal one?

Now we use a consistent word "longitudinal channel profile" (pg. 17 lines 4, 9, 15, caption of Fig. 10). This is the profile in the section between points A and G (pg. 17 line 4). In order to focus on the longitudinal profile, we removed statements on the cross-sectional profile in this section. At the same

time, we renamed points along the longitudinal profile.

• Table 3: the gradient are in degrees. If so please add the "°"

Left tables are standard deviation of slope gradient. Therefore, we have added "(°)" as suggested by the reviewer (Table 3).

If the the authors solved these problems in my opinion the paper will be a nice contribution to the scientific debate on debris-flow initiation.

Thank you again for your comments improving our manuscript.

We sincerely thank you for the efforts you have made to improve our submission to *Natural Hazards and Earth System Sciences*. We have responded to all review comments and have made appropriate modifications to our manuscript related to these comments as detailed in the following paragraphs. The blue-highlighted sentences are the review comments; sentences in black represent our responses to these review comments. As we replied to the first comment, we have removed many sentences that were not directly related to core findings in this manuscript.

The paper may reveal some valuable facts related to the initiation of debris flows in the upper stream. The long-term observation data are useful for understanding the initiation mechanism of debris flows. But the core findings are covered by too many trivial details. The authors use too many words to describe the background information. I think the manuscript could be accepted by the journal if major revisions are made, especially reducing some of unimportant content.

Thank you for reviewing our manuscript. We have received similar comments from the other reviewer. We have removed many sentences that were not closely related to core findings in this manuscript (pg. 11, line 18; pg. 15, line 8; pg. 15, line 9; pg. 15, line 11; pg. 17, line 17; pg. 22, line 4, pg. 22, line 7; pg. 22, line 14; pg. 23, line 8; pg. 23, line 10). Location of these sentences are highlighted green in the revised manuscript. We think structure of the revised manuscript is simpler and clearer compared to previous version of the manuscript.

Specific Comments

1. I think the title is not appropriate, "Interactions between the accumulation of sediment storage and debris flow characteristics in a debris-flow initiation zone, Ohya landslide body, Japan". The paper did not present the "interactions". May be connections or relationship is better.

As suggested by the reviewer, we have replaced "interaction" with "relationships" in the title. We also replaced "interactions" in the text with "relationships" (pg. 1, line 16; pg. 3, line 6; pg. 24, lines 2, 21).

2. Page3 Line 6: "Similarly, shear stress needs to exceed shear strength at the bottom of a traveling sediment mass for the continuity of travel."

Once the debris mass initiates, the shear stress need to keep it moving is equal to or even less than the shear strength sometimes. I think the sentence could be revised.

As the reviewer points out, shear strength in dynamic state is sometimes lower than that in static state. In such cases, shear stress needed for continuity of travel is lower than that needed for initial movement. We have revised the sentence (pg. 3, lines 16-18).

3. The angle is denoted with theta in figure 1a, but it is alpha in the text.

We have replaced θ with α in the figure (Fig. 1).

4. It is better if the images in figures such as Figure 3 have a scale.

As suggested by the reviewer, we have added a scale in Fig 3a.

5. 4.1 and 4.2 sections are too long. I think a brief description of the monitoring and TLS measuring can make readers focus on the topic.

We have shortened length of sentences to make the section shorter (pg. 8, lines 4, 13, 16; pg.10, lines 8). Current length of the sections may be still long for the reviewer. However, many of information in these sections were requested by the other reviewers in previous stage of the review process. We think such information should not be removed. Instead, we simplified section 4.3 based on the comment by another reviewer. Total length of chapter 4 in the revised manuscript is much shorter than that in the previous version of the manuscript.

6. Some discussions should be cautious. For example, Page 20 Line 21: "The proportional duration of partly saturated flow in the overall surges had a positive relationship with the volume of storage (Fig. 6). Thus, the volume of storage was a factor controlling not only the initiation of the debris flow, particularly in the supply limited basin (Bovis and Jakob, 1999; Jakob et al., 2005), but also the debris flow characteristics."

I think the fact of "The proportional duration of partly saturated flow in the overall surges had a positive relationship with the volume of storage" cannot deduce the inevitable conclusion. I suggest "It indicates that" in place of "Thus".

As the reviewer points out, the relationship between flow type and volume of storage may not strongly support the idea that the volume of storage control flow characteristics. We replaced "thus" with "it indicates that" (pg. 21, line 21). We have carefully checked all expressions in the discussion again, and revised sentences which were not appropriate (pg. 21, line 30, 31; pg.22, lines 6-7; pg. 23, line 11)

[revised manuscript text omitted]

---

## Author Response (AR3)

**Reply for review comments**

**Reviewer 1 (Prof. Carlo Gregoretti)**

We sincerely thank you for the efforts you have made to improve our submission to *Natural Hazards and Earth System Sciences*. We have responded to all review comments and have made appropriate modifications to our manuscript related to these comments as detailed in the following paragraphs. The blue-highlighted sentences are the review comments; sentences in black represent our responses to these review comments.

Overview
The authors improved their work but there are some parts to be refined:
1) The subsection 4.3 is not understandable. The first sentence is not clear: what does it mean "We estimated the volume of storage from periodical photography and terrains …". For the following part, I suggest the authors to write that 36 cross section lines were considered along the reach and volume of storage was computed for each of the 35 areas between the cross-section line. Moreover, at the beginning of the second sentence did the authors confuse storage for channel?

We revised the first and second sentences in the section to improve understanding by readers. We computed "cross-sectional area" of the storage at cross-sectional lines, and did not computed areas between the sections. We have checked the sentence with a native English speaker again (pg. 11, lines 13-17).

2) About debris flow monitoring (5.1), the reviewer does not understand the different separation time of rainfall durations for the two thresholds, 10 h, when the rainfalls are classified in two groups based on rainfall duration smaller or larger than 5h. Moreover, legend of Figure 5 should be corrected: no debris flow instead of debirs flow. The unknown points of Figure 5 need some specification or explanation in the caption or in the text: do they correspond to fully saturated debris flows or both to partly and fully unsaturated debris flows? About the weak positive dependence of partly saturated debris flow on the storage volume, a reason could be the dependence from another factor: the precipitated depth before debris flow occurrence (see point 4).

As shown in Figure 5, there are few debris flow events with rainfall duration of about 5 h, indicating that most of rainfall events can be grouped into two groups: long-lasting rainfall events caused by typhoons and stationary fronts (rainfall duration >5 h), and short-duration convective rainfall events characterized by high intensity (rainfall duration <5 h). Rainfall threshold lines in the Figure 5 express initiation condition of the debris flow, and was not used to classification of rainfall events.

As pointed out by the reviewer, we have corrected the misspelling "debirs flow" (Figure 5, legend).

We have added explanation on the unknown points in the figure caption. Because we failed to capture debris flow images during these "unknown" events, we do not have any information on the flow type. As pointed out by the reviewer, precipitation depth could be another factor affecting debris flow occurrence. However, we could not find clear relationship between rainfall depth and occurence of debris flows (see next figure

showing comparison between rainfall depth and rainfall-intensity). Effect of long-time rainfall factors, including rainfall depth before debris flow occurrence, on the occurrence of debris flow is smaller than that of short-time rainfall intensity.

[Figure]

3) About subsection 5.3, where are the points F and G on Figure 2?

Although we have renamed points in the last revision, we forgot to revise point name in the Figure 2. We have revised the Figure 2.

4) About section 6.1, the sentence at the lines 16-19 of page 21 is not very clear: could you rewrite? Moreover, the writer agrees with the authors that the storage volume can influence the debris flow type. About the two debris flows occurred at Cancia in summer of year 2015 (Gregoretti et al., 2016), the former was partly saturated in the first minute while the latter was fully saturated. The reason is that the upstream storage volume in the second case was negligible because washed out by the first event occurred 12 days before. Authors, if willing, could insert this fact in this subsection. Considering Figure 7, another factor controlling the debris flow type could be the rainfall precipitated before the debris flow occurrence in the case of long duration rainfalls (typhoons). In this case the storage volume could be partly or entirely saturated by the rainfall previous debris flow and the degree of saturation could influence the proportional amount of partly saturated debris flow. Authors could explore this possibility after analyzing the rainfall data.

We have rewritten the sentence at the lines 15-18 in pg. 21. The sentence is now "Similarly, debris flow initiation zones in many other torrents are greater than 22.2° (e.g., VanDine 1985; McCoy et al., 2012). Although the $\alpha_2$ may be different among torrents affected by soil parameters such as the internal angle of friction, the conditions for the occurrence of partly saturated flow may possibly be satisfied in such debris flow initiation zones." We also introduced implication in Gregoretti et al., 2016 to support our discussion (pg. 21, lines 20-21).

We agree that the precipitation before debris flow events is the potential factor affecting the flow type. At the

same time, effect of the precipitation before the debris flow event would be vary affected by the volume of the storage. Large space would be needed to complete discussion on such initial water condition in the storage. We would like to analyze the impact of previous rainfall events on the debris flow type in future papers.

5) In section 7, authors should add some specification to support the sentence "In addition, our study elucidated that the slope geomorphic units is the key factor in the estimation of predominant type of the sediment transport…"

Based on the suggestion by the reviewer, we have added "talus slopes and channel deposits" in the sentence (pg. 24, line 13).

The following are the detailed comments and specifications.
1. Page 5 – line 9: ratio instead of ration?

We have replaced "ration" to "ratio" (pg. 5, line 9).

2. Page 22- line 27: what is it the length ratio of the channel sections? (see also Figure 11)

We have replaced "length ratio" to "proportional amount" (pg. 22, lines 26-28). We also replaced "length ratio" to "proportional amount" in y-axis and figure caption of Figure 11. Calculation method of the proportional amount is explained in pg. 20, line 11 –pg. 21 line 9.

3. Page 23– line 20: please insert could before exist.

We have inserted "could" as suggested by the reviewer (pg. 23, line 19).

4. Page 24 - line 27: is instead of are after Paolo Tarolli.

We have replaced "are" by "is" (pg. 24, line 24).